# Adipocyte HIF2α functions as a thermostat via PKA Cα regulation in beige adipocytes

Ji Seul Han [1], Yong Geun Jeon[1], Minsik Oh[2], Gung Lee[1], Hahn Nahmgoong[1], Sang Mun Han[1], Jeehye Choi[1], Ye Young Kim[1], Kyung Cheul Shin[1], Jiwon Kim[1], Kyuri Jo [3], Sung Sik Choe[1], Eun Jung Park[4], Sun Kim[5,6,7] & Jae Bum Kim [1]✉

Thermogenic adipocytes generate heat to maintain body temperature against hypothermia in response to cold. Although tight regulation of thermogenesis is required to prevent energy sources depletion, the molecular details that tune thermogenesis are not thoroughly understood. Here, we demonstrate that adipocyte hypoxia-inducible factor α (HIFα) plays a key role in calibrating thermogenic function upon cold and re-warming. In beige adipocytes, HIFα attenuates protein kinase A (PKA) activity, leading to suppression of thermogenic activity. Mechanistically, HIF2α suppresses PKA activity by inducing miR-3085-3p expression to downregulate PKA catalytic subunit α (PKA Cα). Ablation of adipocyte HIF2α stimulates retention of beige adipocytes, accompanied by increased PKA Cα during re-warming after cold stimuli. Moreover, administration of miR-3085-3p promotes beige-to-white transition via downregulation of PKA Cα and mitochondrial abundance in adipocyte HIF2α deficient mice. Collectively, these findings suggest that HIF2α-dependent PKA regulation plays an important role as a thermostat through dynamic remodeling of beige adipocytes.

[1] Center for Adipocyte Structure and Function, Institute of Molecular Biology and Genetics, School of Biological Sciences, Seoul National University, Seoul, South Korea. [2] School of Software Convergence, Myongji University, Seoul, South Korea. [3] Department of Computer Engineering, Chungbuk National University, Cheongju, South Korea. [4] Cancer Immunology Branch, Department of System Cancer Science, National Cancer Center, Goyang, South Korea. [5] Department of Computer Science and Engineering, Seoul National University, Seoul, South Korea. [6] Institute of Engineering Research, Seoul National University, Seoul, South Korea. [7] AIGENDRUG Co., Ltd., Seoul, South Korea. ✉email: jaebkim@snu.ac.kr

Adipose tissue is a central organ in the regulation of energy homeostasis. In mammals, adipose tissues are largely divided into white adipose tissue (WAT) and brown adipose tissue (BAT) based on their morphology and functional characteristics. While WAT is responsible for lipid storage and endocrine functions, BAT is specialized in the maintenance of body temperature by non-shivering thermogenesis to protect the body from hypothermia[1]. Brown adipocytes in BAT contain multilocular lipid droplets (LDs) and numerous mitochondria, which are suitable for efficient heat generation by dissipating proton gradient via uncoupling protein 1 (UCP1). Recently, it has been reported that WAT is able to possess brown-like adipocytes, termed beige or brite adipocytes[2,3]. As beige adipocytes exhibit morphological and functional plasticity upon stimuli, they are considered to be inducible types of thermogenic adipocytes[4–6]. Upon demands for long-term heat production such as chronic cold exposure or sustained β-adrenergic activation, enhanced beige adipocytes stimulate the thermogenic programs and increase UCP1 expression. In contrast, beige adipocytes lose their thermogenic characteristics and return to white adipocytes accompanied by unilocular LD and decreased mitochondrial abundance upon re-warming or alteration of external stimuli[5,7,8].

Upon sympathetic nerve activation during cold acclimation, activation of β-adrenergic receptor-dependent adenylyl cyclase produces cyclic AMP (cAMP), which in turn stimulates protein kinase A (PKA) signaling. In thermogenic adipocytes, PKA signaling promotes lipolysis through phosphorylation of lipases and related proteins to produce free fatty acids and glycerol[9–11]. The fatty acids generated from PKA-induced lipolysis are not only used as a fuel but also directly activate UCP1 in mitochondria[12]. Besides, PKA governs the activation of thermogenic programming including thermogenic gene expression and mitochondrial biogenesis[13–15]. Although PKA activation largely depends on cellular cAMP levels, emerging evidence has proposed that the stoichiometric balance between PKA regulatory and catalytic subunits is also a key factor in modulating PKA activity, independent of cAMP levels[16–18].

Thermogenic adipocytes utilize large amounts of energy source and oxygen to fulfill their needs for sustaining their activities under cold conditions. As increased oxygen consumption leads to an oxygen deficit in adipose tissues, thermogenic adipose tissues become hypoxic upon cold exposure[19]. Under oxygen deprivation, hypoxia-inducible factor α (HIFα) escapes from prolyl hydroxylase (PHD)- and von Hippel–Lindau protein (VHL)-dependent proteasomal degradation and modulates various cellular responses, including glucose and lipid metabolism[20–22]. HIFα has two major α subunits, HIF1α and HIF2α, which regulate common and specific target genes[20]. During the catabolic process, HIFα affects lipid metabolism by modulating lipolysis and mitochondrial fatty acid oxidation[23,24]. In adipocytes, HIFα attenuates PKA-induced lipolysis to prevent futile lipid breakdown[25]. Moreover, HIFα has been reported to suppress mitochondrial activities including β-oxidation and oxidative phosphorylation (OXPHOS) in hypoxic conditions and VHL deficient models[26–29]. Recently, it has been reported that fat mass and adipocyte size are altered in adipocyte-specific HIF1α or HIF2α knockout mice upon high fat diet[30–32]. However, the physiological roles of HIFα in cold-induced hypoxia and, particularly, the key isoform of HIFα in thermogenic adipocytes have not been thoroughly studied.

In this study, we aimed to unravel the roles of HIFα in thermogenic adipocytes. We comprehensively analyzed the thermoregulatory roles of HIFα using various adipocyte-specific HIFα knockout mouse models and pharmacological HIFα modulation. In addition, bioinformatic approaches were adopted to identify the key mediator(s) of HIF2α-dependent pathways for PKA-

dependent thermogenic regulation. Here, we elucidate the physiological roles of HIF2α in the activation of beige adipocytes and rewiring of beige-to-white adipocyte transition. Collectively, our data suggest that HIF2α is an important molecular brake that fine-tunes thermogenic activity in beige adipocytes to maintain whole-body energy homeostasis.

## Results

**Cold acclimation increases HIFα expression in thermogenic adipocytes.** Given that HIFα could be induced in thermogenic adipose tissues[19], we evaluated the extent of hypoxia in several fat depots at different temperatures. Compared to thermoneutral (TN) conditions, the overall intensity of pimonidazole staining was markedly increased in inguinal WAT (iWAT) and BAT during cold exposure (Fig. 1a, b), but not in epididymal WAT (eWAT) (Supplementary Fig. 1a). Simultaneously, the levels of HIF1α and HIF2α proteins gradually increased during cold exposure in iWAT and BAT (Fig. 1c, d), and HIF2α proteins were slightly induced in eWAT upon cold (Supplementary Fig. 1b). Moreover, in iWAT, cold-induced HIF1α and HIF2α protein levels were augmented in adipocytes with multilocular LDs (Fig. 1e, f). To examine whether HIFα might be also promoted by β-adrenergic stimuli, we measured the levels of HIF1α and HIF2α protein upon treatment with a β3-adrenergic agonist, CL-316,243 (CL). In iWAT and BAT, chronic administration of CL elevated HIF1α and HIF2α as well as UCP1 (Fig. 1g, h). These data indicate that HIFα protein expression would be augmented in thermogenic adipose tissues, accompanied by hypoxia.

**HIFα regulates thermogenic programming in adipocytes upon cold stimuli.** To investigate the roles of HIFα in thermogenic adipocytes, we generated adipocyte-specific HIF1α, HIF2α knockout and HIF1/2α double knockout (HIF1α AKO, HIF2α AKO, and HIF1/2α DKO, respectively) mice using *Adiponectin-Cre*. As expected, mRNA levels of the *Hif1a* and/or *Hif2a* gene were downregulated in adipose tissue (Supplementary Fig. 2a). Upon cold exposure, HIF1α AKO, HIF2α AKO, and HIF1/2α DKO mice were cold tolerant and exhibited higher body temperature than wild-type (WT) mice (Fig. 2a, b). Next, we subjected the thermogenic tissues to histological analyses. While BAT did not show significant differences in WT and adipocyte HIFα deficient mice upon TN or cold (Supplementary Fig. 2b–d), multilocular LDs containing adipocytes were evidently elevated in iWAT of HIF1αAKO, HIF2α AKO, and HIF1/2α DKO mice upon cold (Fig. 2c–e). Thermogenic gene expression was significantly upregulated in iWAT of HIF2α AKO and HIF1/2α DKO mice and slightly increased in iWAT HIF1α AKO mice (Fig. 2f), implying that HIF2α might be a key player in the regulation of thermogenic programming. Consistently, the level of UCP1 protein was further promoted in cold exposed iWAT of HIF2α AKO mice than that of WT mice (Supplementary Fig. 2e, f). Accordingly, HIFα deficient beige adipocytes potentiated thermogenic gene expression, while overexpression of HIFα attenuated thermogenic gene expression (Supplementary Fig. 2g, h).

To explore whether HIFα could modulate thermogenic function in classical brown adipocytes, we generated brown adipocyte-specific HIF1α and HIF2α knockout (HIF1α BKO and HIF2α BKO, respectively) mice using *Ucp1-Cre*. Similar to the pan-adipocyte HIFα deletion mouse model, HIF1α BKO and HIF2α BKO mice were cold tolerant (Supplementary Fig. 3a, b). While the overall size of lipid droplets was relatively smaller in BAT of HIF1α BKO and HIF2α BKO than that of WT at 6 h of cold (Supplementary Fig. 3c), histological changes appeared to be comparable between the genotypes upon 3 days of cold (Supplementary Fig. 3d), implying that thermogenic function

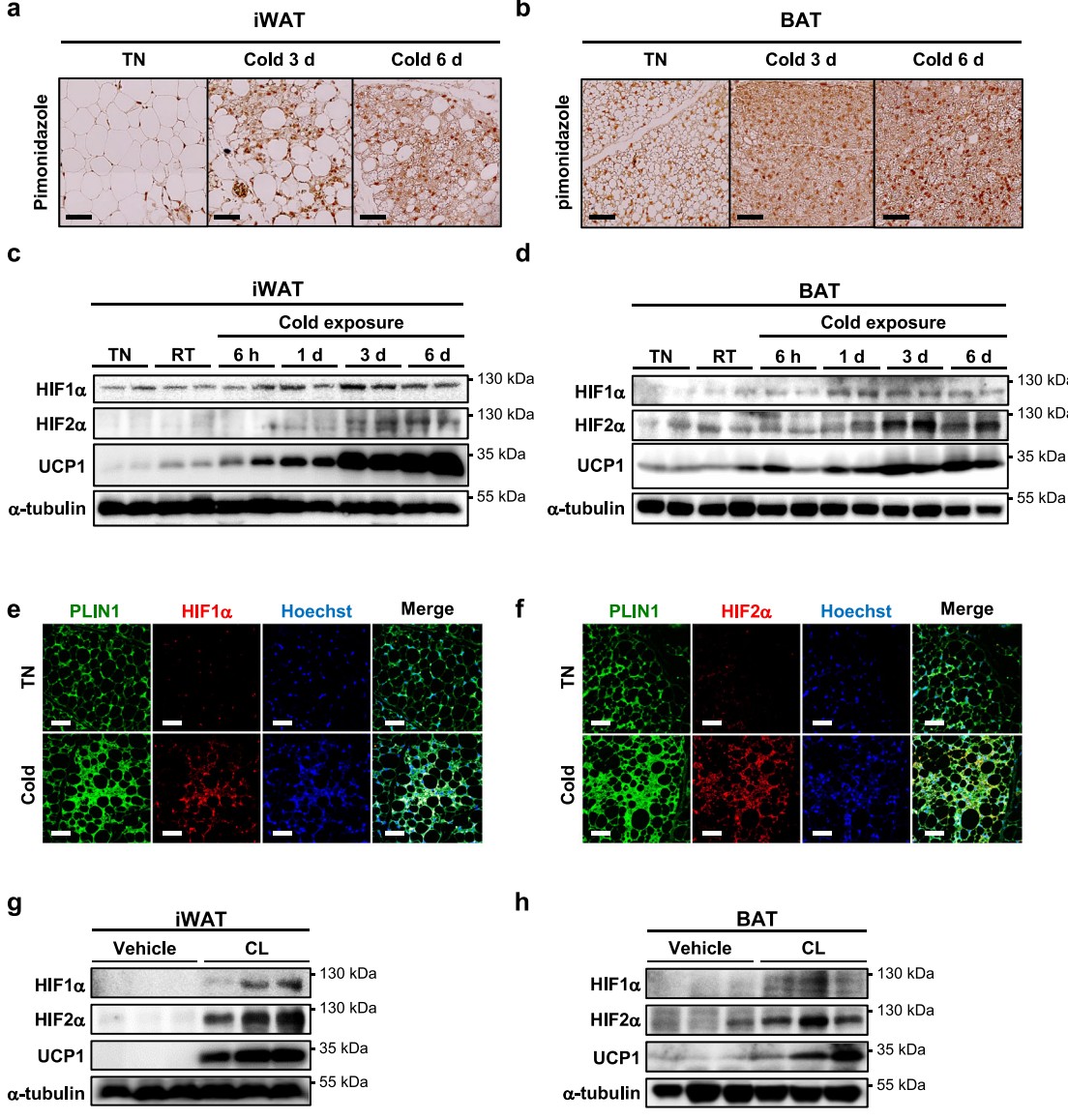

**Fig. 1 Cold exposure induces HIFα expression in thermogenic adipose tissues. a**, **b** Immunohistochemistry of (**a**) iWAT and (**b**) BAT sections upon TN or cold exposure using an anti-pimonidazole antibody and DAB staining. Scale bars, 50 μm. **c**, **d** Western blot analysis of HIFα and UCP1 (loaded 50 μg for iWAT and 10 μg for BAT) in **c** iWAT and **d** BAT upon TN (30 °C), RT (22 °C), or cold exposure (4-6 °C). **e**, **f** Immunofluorescence images of iWAT sections upon TN or cold exposure (3 days) using anti-PLIN1 (green) and **e** anti-HIF1α or **f** anti-HIF2α (red) antibodies with Hoechst (blue) staining. Scale bars, 50 μm. **g**, **h** Western blot analysis of HIFα and UCP1 in **g** iWAT and **h** BAT at TN upon daily CL administration (0.5 mg/kg, 4 days).

might be maximized by 3 days of cold exposure. In addition, many thermogenic gene expressions were comparable in BAT and iWAT, except that *Ucp1* and *Elovl3* appeared to be upregulated in BAT of HIF2α BKO mice (Supplementary Fig. 3e, f).

In parallel, thermogenic properties were investigated by pharmacological interventions of HIFα with a general HIFα inhibitor (YC-1), HIF2α-specific inhibitor (PT2385), or HIFα activator (dimethyloxaloylglycine, DMOG) during cold acclimation (Supplementary Fig. 4a–f). Similar to the genetic ablation of HIFα, YC-1-treated WT mice were cold tolerant with increased thermogenic gene expression and beige adipocyte accumulation in iWAT upon cold (Fig. 2g–i). In addition, HIF2α-specific inhibitor PT2385-treated mice exhibited augmented thermogenic properties with elevated beige adipocytes in iWAT upon cold (Fig. 2j–l and Supplementary Fig. 4g). On the contrary, DMOG-administered WT mice were cold intolerant with attenuated thermogenic gene expression and beige adipocyte formation in iWAT upon cold (Fig. 2m–o). Therefore, these data suggest that

adipocyte HIFα inhibition could stimulate thermogenic programming via beige adipocyte generation during cold exposure.

**Loss of HIF2α in adipocytes increases energy expenditure upon β-adrenergic activation.** As chronic β-adrenergic stimulation potentiates beige adipocyte formation in iWAT[33], we examined the effects of chronic CL treatment on thermogenic activity in iWAT. As shown in Fig. 3a–c, iWAT of HIF2α AKO mice exhibited elevated thermogenic properties as well as increased UCP1 expression upon CL treatment. Similarly, CL boosted beige adipocyte generation and thermogenic gene expression in HIF1α AKO and HIF1/2α DKO iWAT (Supplementary Fig. 5a–d). To study whether increased thermogenic activity by adipocyte HIFα ablation might alter whole-body energy metabolism, metabolic rates were measured. As indicated in Fig. 3d–i and Supplementary Fig. 5e–g, the volume of oxygen consumption ($VO_2$), the volume of carbon dioxide production ($VCO_2$), and energy expenditure

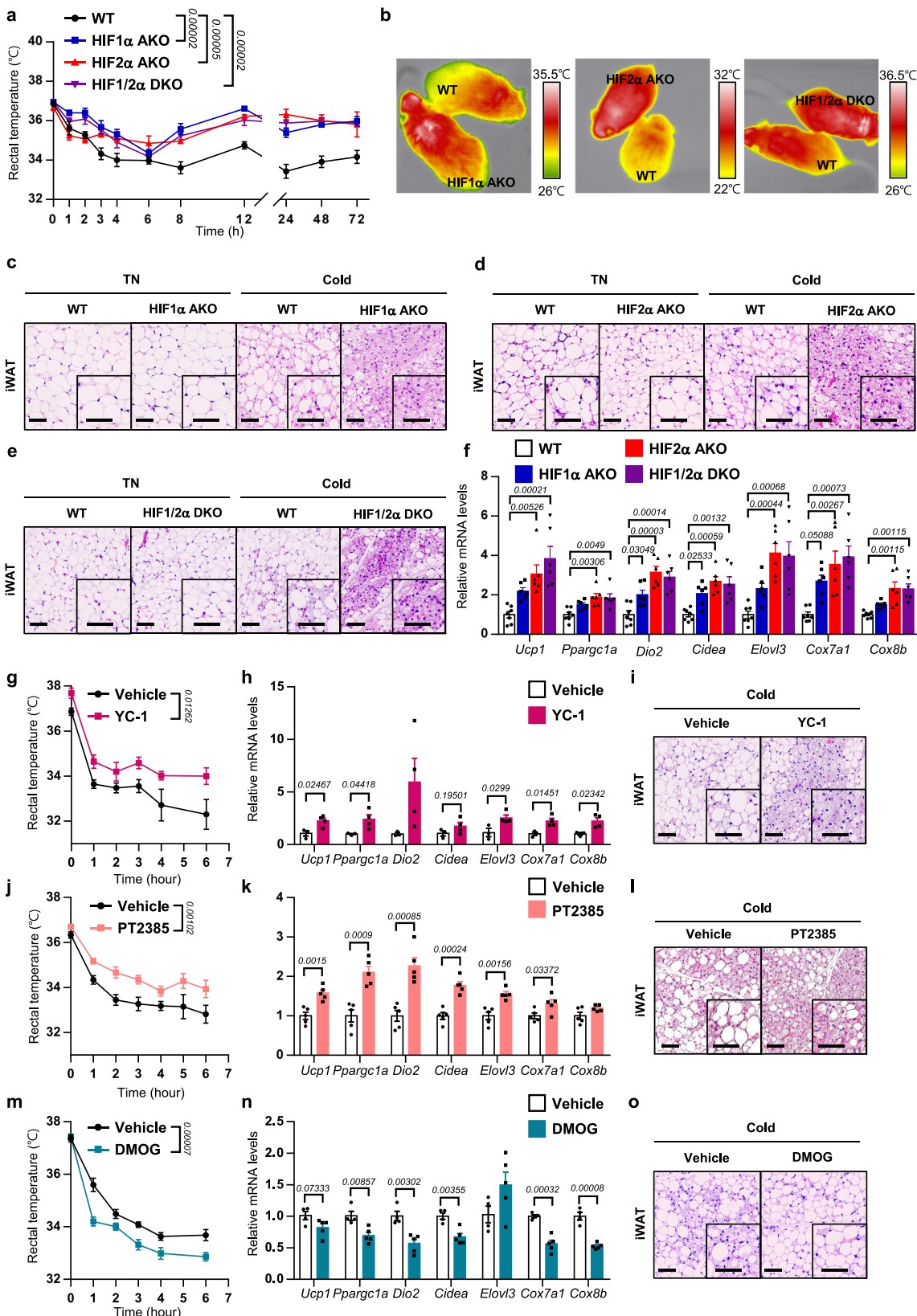

were increased in HIF2α AKO and HIF1/2α DKO mice treated with CL. However, these parameters were comparable between WT and HIF1α AKO mice treated with CL (Supplementary Fig. 5h–j), implying that adipocyte HIF2α deficiency would potentiate energy expenditure with β-adrenergic activation.

**In beige adipocytes, HIF2α ablation potentiates PKA signaling via PKA Cα upregulation.** To decipher the underlying mechanism(s) by which HIF2α modulates thermogenesis, transcriptome profiles of iWAT were scrutinized. Compared to WT iWAT, HIF2α AKO iWAT showed numerous differentially

**Fig. 2 Adipocyte-specific HIF2α KO mice exhibit increased thermogenic activities upon cold exposure. a** Changes in rectal temperature of WT ($n = 9$), HIF1α AKO ($n = 5$), HIF2α AKO ($n = 6$), and HIF1/2α DKO ($n = 7$) mice during cold exposure. **b** Infrared images of body surface temperature of WT, HIF1α AKO, HIF2α AKO, and HIF1/2α DKO mice upon cold exposure (4 h). **c–e** Representative images of hematoxylin and eosin (H&E) staining of iWAT from WT, **c** HIF1α AKO, **d** HIF2α AKO, and **e** HIF1/2α DKO mice upon TN or cold exposure (3 days). Scale bars, 50 μm. **f** mRNA levels in iWAT from WT ($n = 7$), HIF1α AKO ($n = 6$), HIF2α AKO ($n = 6$), and HIF1/2α DKO ($n = 6$) mice upon cold exposure (3 days). **g** Rectal temperature of vehicle- ($n = 5$) and YC-1- (30 mg/kg, $n = 5$) administered mice during cold exposure. YC-1 was injected i.p. 1 h prior to exposure to cold. **h** mRNA levels in iWAT from vehicle- ($n = 3$) and YC-1- ($n = 4$) administered mice upon cold exposure (3 days). **i** Representative images of H&E staining of iWAT from vehicle- and YC-1-administered mice upon cold exposure (3 days). Scale bars, 50 μm. **j** Rectal temperature of vehicle- ($n = 6$) and PT2385- (10 mg/kg, $n = 6$) administered mice during cold exposure. PT2385 was injected i.p. 1 h prior to exposure to cold. **k** mRNA levels in iWAT from vehicle- ($n = 5$) and PT2385- ($n = 5$) administered mice upon cold exposure (3 days). **l** Representative images of H&E staining of iWAT from vehicle- and PT2385-administered mice upon cold exposure (3 days). Scale bars, 50 μm. **m** Rectal temperature of vehicle- ($n = 6$) and DMOG- (40 mg/kg, $n = 7$) administered mice during cold exposure. DMOG was injected i.p. 1 h prior to exposure to cold. **n** mRNA levels in iWAT from vehicle- ($n = 4$) and DMOG- ($n = 5$) administered mice upon cold exposure (3 days). **o** Representative images of H&E staining of iWAT from vehicle- and DMOG-administered mice upon cold exposure (3 days). Scale bars, 50 μm. Data were expressed as the mean ± SEM by either two-tailed unpaired Student $t$-tests in (**h**, **k**, **n**), one-way ANOVA in (**f**), or two-way repeated-measures ANOVA in (**a**, **g**, **j**, **m**) followed by Holm–Sidak's multiple comparisons test.

expressed genes (DEGs) (Supplementary Fig. 6a). Among the upregulated DEGs, "thermogenesis" and "mitochondrion" gene ontology signatures were ranked as the top biological pathways in cold exposed iWAT of HIF2α AKO mice (Fig. 4a, b). To investigate the key mediator(s) influencing DEGs in thermogenesis and related pathways, in silico analyses were conducted. Network propagation (NP) scores of genes were calculated to measure their relevance to HIF2α (Supplementary Fig. 6b and Supplementary Data 1) and the centrality of genes was calculated to measure their importance in the network (Supplementary Fig. 6c). Among several candidates, it was of interest to note that *Prkaca*, which encodes PKA catalytic subunit α (PKA Cα), was one of the highly ranked genes which could influence numerous catabolic pathways including PKA-related, TCA cycle, fatty acid β-oxidation, OXPHOS, and thermogenesis in iWAT of HIF2α AKO mice upon cold (Fig. 4c, d).

Next, we examined whether PKA Cα expression and PKA signaling might be indeed altered by HIF2α. As shown in Fig. 4e–g, PKA Cα was elevated in iWAT of HIF2α AKO mice upon cold or CL treatment. Also, the levels of phosphorylated PKA target proteins were further augmented in iWAT and beige adipocytes of HIF2α AKO mice than those of WT mice upon cold or β-adrenergic activation (Fig. 4h–j). Additionally, the levels of serum glycerol and glycerol released from beige adipocytes were enhanced in HIF2α deficient adipocytes upon cold or ISO treatment (Supplementary Fig. 6d and Fig. 4k). Nevertheless, there was no significant difference in the cAMP level between WT and HIF2α deficient beige adipocytes (Fig. 4l), implying that HIF2α could regulate PKA activity independent of cAMP. To ascertain whether PKA Cα would be regulated by HIF2α in beige adipocytes, the levels of PKA Cα expression were examined with HIF2α modulation. PKA Cα protein was upregulated in HIF2α deficient beige adipocytes (Fig. 4m, n), whereas overexpression of HIF2α attenuated PKA Cα expression (Fig. 4o, p). Similar to HIF2α AKO iWAT, HIF1α AKO and HIF1/2α DKO iWAT also exhibited enhanced PKA signaling and PKA-induced lipolysis (Supplementary Fig. 6e–g). However, *Prkaca* expression was not altered by HIF1α modulation (Fig. 4e and Supplementary Fig. 6h–k). Together, these results suggest that HIFα suppresses PKA signaling in beige adipocytes, and PKA Cα expression is selectively regulated by HIF2α.

**In beige adipocytes, HIF2α ablation stimulates mitochondrial activity.** The finding that "mitochondrion" was top-ranked in iWAT of HIF2α AKO mice upon cold (Fig. 4b) led us to test whether mitochondrial activity might be influenced by HIF2α. As shown in Fig. 5a–c, mtDNA and OXPHOS complexes were increased in HIF2α AKO iWAT upon cold. Similarly, chronic CL

administration augmented mitochondrial quantity in HIF2α AKO iWAT (Fig. 5d–f). In contrast, in beige adipocytes, HIF2α overexpression attenuated mtDNA content, whereas HIF2α deficiency increased mtDNA content (Supplementary Fig. 7a, b). Similar to iWAT in HIF2α AKO, mitochondrial OXPHOS complexes were also upregulated in HIF1α AKO iWAT upon cold (Supplementary Fig. 7c, d). Next, to study whether the altered mitochondrial content in HIF2α AKO iWAT might affect mitochondrial activity, mitochondrial oxygen consumption rates (OCRs) were determined. OCRs in basal, UCP1-dependent, and maximal states were repressed by overexpression of HIF2α (Fig. 5g). In contrast, HIF2α deficient beige adipocytes exhibited increased basal, UCP1-dependent, and maximal respiration (Fig. 5h), with an increase in mitochondrial membrane potential (Supplementary Fig. 7e). Further, we found that enhanced mitochondrial respiration, as well as upregulated thermogenic gene expression in HIF2α deficient beige adipocytes, was impaired by PKA inhibitor H89 treatment or knockdown of *Prkaca* (Fig. 5i and Supplementary Fig. 7f–h). To further elucidate in vivo roles of PKA Cα in iWAT of HIF2α AKO mice, si*Prkaca* was delivered to iWAT (Fig. 5j). Suppression of PKA Cα via siRNA downregulated UCP1 expression, mitochondrial OXPHOS complexes, accompanied with attenuated beige adipocytes formation in iWAT of HIF2α AKO mice upon cold (Fig. 5k, l). These data propose that HIF2α deficiency could result in elevated mitochondrial content and mitochondrial activity in beige adipocytes.

**Adipocyte HIF2α suppresses thermogenesis via miR-3085-3p-dependent PKA Cα repression.** To delineate how HIF2α modulates *Prkaca*, we assessed DEGs between WT and HIF2α AKO mice. Although several DEGs in HIF2α AKO iWAT were identified, it appeared that there were no candidate genes which could directly repress transcription of *Prkaca* (Supplementary Data 2). Since HIFα could suppress expression of certain genes via induction of miRNA[34], we hypothesized that HIF2α-dependent miRNA expression might mediate the downregulation of *Prkaca*. Using in silico analysis, we identified miR-3085-3p as one of the highest-scoring miRNAs targeting the evolutionarily conserved region of 3′UTR of *Prkaca*. In a reporter assay, miR-3085-3p attenuated luciferase activity of WT *Prkaca* 3′UTR, but not that of the miR-3085-3p binding defective mutant of *Prkaca* 3′UTR (Fig. 6a, b). To evaluate whether HIF2α would be recruited to stimulate miR-3085-3p expression, ChIP-qPCR analysis was performed. As shown in Fig. 6c, d and Supplementary Fig. 8a, overexpressed HIF2α interacted with several hypoxia-response elements in the upstream region of miR-3085. Moreover, the level of miR-3085-3p was regulated in a HIF2α-dependent manner in

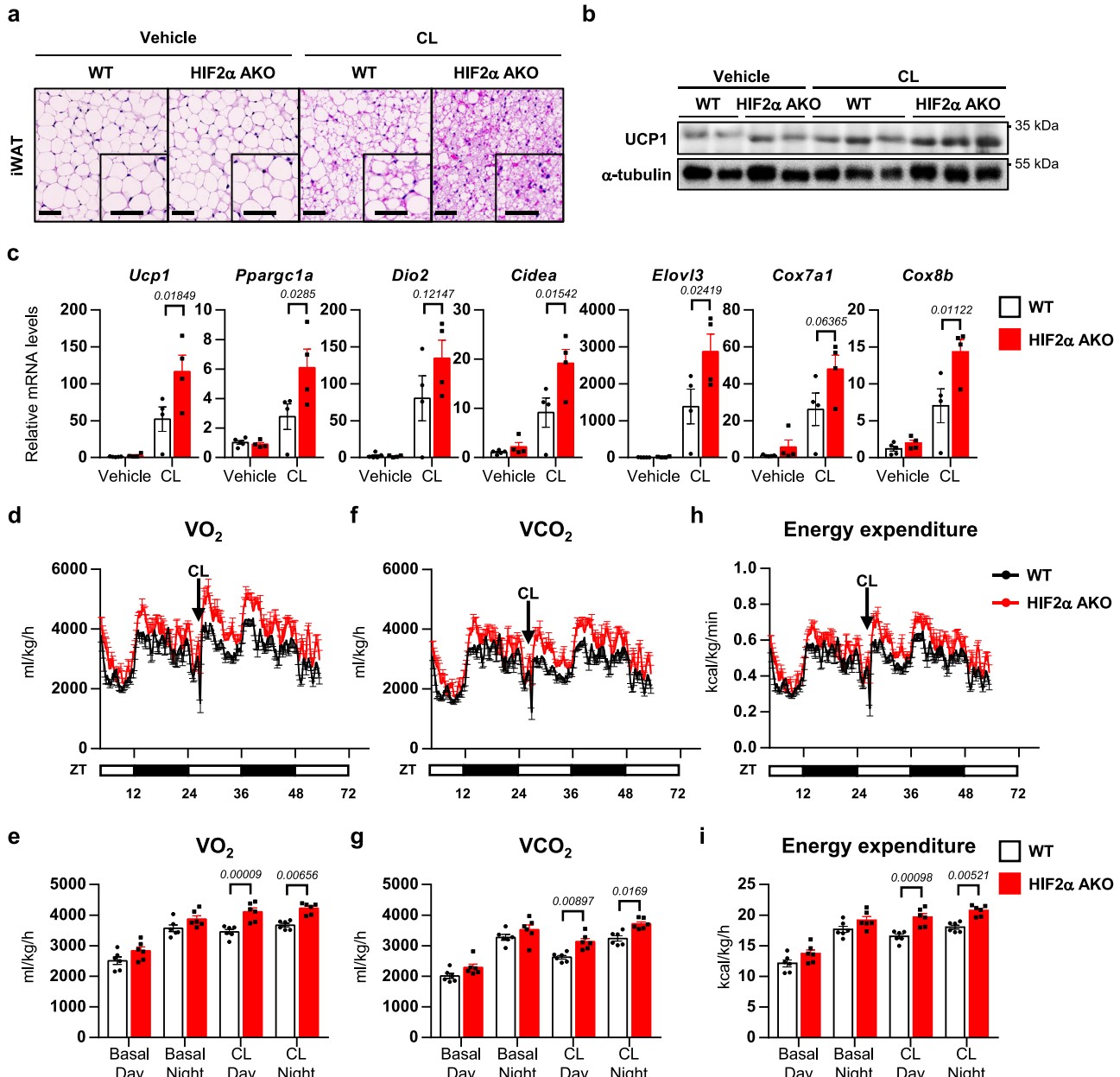

**Fig. 3 HIF2α ablation in adipocytes potentiates energy expenditure upon β3-adrenergic activation. a** Representative images of H&E staining of iWAT from WT and HIF2α AKO mice upon daily CL administration (0.5 mg/kg, 4 days). Scale bars, 50 µm. **b** Western blot analysis of UCP1 in iWAT from WT and HIF2α AKO mice upon daily CL administration (0.5 mg/kg, 4 days). **c** mRNA levels in iWAT from WT (Vehicle; $n = 5$, CL; $n = 4$) and HIF2α AKO ($n = 4$) mice upon daily CL administration (0.5 mg/kg, 4 days). **d**–i **d**, **e** VO$_2$, **f**, **g** VCO$_2$, and **h**, **i** energy expenditure for WT and HIF2α AKO mice ($n = 6$). Data were expressed as the mean ± SEM by two-way ANOVA followed by Holm–Sidak's multiple comparisons test.

iWAT, BAT, and thermogenic adipocytes (Fig. 6e, f and Supplementary Fig. 8b–i). Consistent with these, HIF2α-specific inhibitor PT2385 decreased miR-3085-3p expression, accompanied by increased *Prkaca* expression in iWAT upon cold exposure (Fig. 6g, h). These data indicate that HIF2α could selectively activate miR-3085-3p expression in thermogenic adipocytes.

Next, we investigated whether miR-3085-3p might function as a downstream mediator of HIF2α in the regulation of PKA Cα and thermogenesis. Increased PKA Cα expression in HIF2α deficient beige adipocytes was potently attenuated by miR-3085-3p mimic, leading to repression of PKA signaling and thermogenic gene expression (Fig. 6i and Supplementary Fig. 8j). Further, miR-3085-3p mimic suppressed the effect of HIF2α

deletion on elevated mitochondrial respiration (Fig. 6j). Conversely, miR-3085-3p inhibitor restored decreased PKA Cα expression, PKA signaling, thermogenic gene expression, and OCRs in HIF2α overexpressing beige adipocytes (Fig. 6k, l and Supplementary Fig. 8k), indicating that miR-3085-3p would be a key mediator of HIF2α-dependent thermogenic regulation. To further examine the in vivo effects of miR-3085-3p in iWAT of HIF2α AKO mice, miR-3085-3p mimic was administered into iWAT (Fig. 6m). As shown in Fig. 6n, o, miR-3085-3p mimic downregulated beige adipocyte generation and UCP1 expression in cold exposed HIF2α AKO iWAT. Similar to beige adipocytes, miR-3085-3p mimic attenuated PKA signaling and thermogenic gene in brown adipocytes (Supplementary Fig. 8l, m). Moreover, administration of miR-3085-3p mimic in BAT suppressed PKA

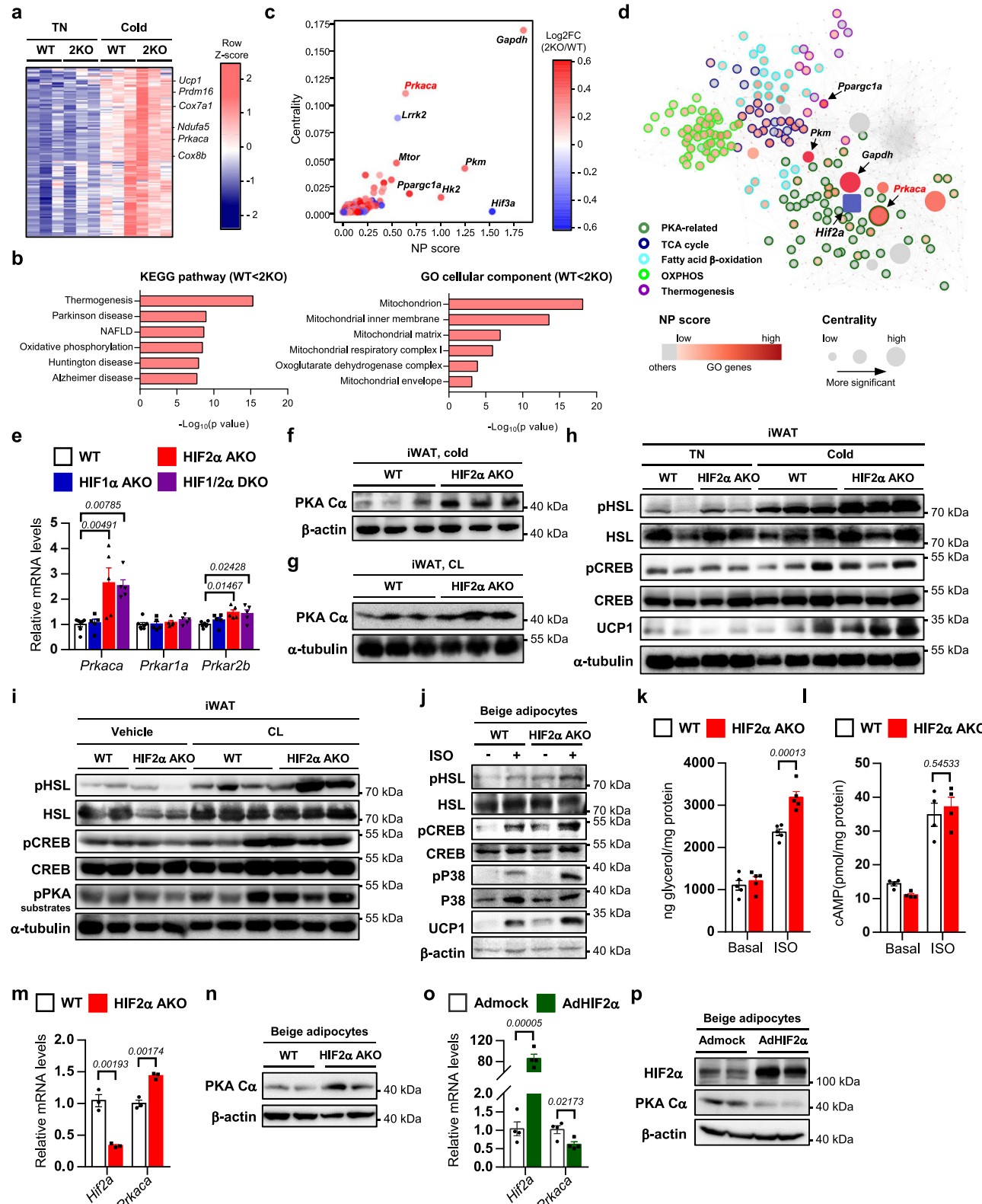

Cα, UCP1, and thermogenic genes (Supplementary Fig. 8n–p) Together, these data suggest that adipocyte HIF2α could promote miR-3085-3p expression, eventually leading to suppression of PKA Cα-mediated thermogenic properties.

**HIF2α confers whitening of beige adipocytes upon re-warming stimuli.** It has been reported that beige adipocytes could regain key features of white adipocytes upon re-warming or removal of

β-adrenergic activation[4,7]. During re-warming, mitochondria in beige adipocytes undergo dynamic remodeling, including down-regulation of mitochondrial function-related gene expression and upregulation of mitochondrial clearance[5,35]. The findings that HIF2α could act as a negative regulator of thermogenesis prompted us to evaluate that HIF2α might be involved in the whitening process of beige adipocytes. As shown in Fig. 7a, HIF2α was sustainably expressed in iWAT upon re-warming. To

**Fig. 4 PKA signaling is activated by an increase in PKA Cα in HIF2α deficient adipocytes. a** Heatmap of upregulated DEGs in iWAT of HIF2α AKO mice upon cold exposure (3 days). **b** KEGG pathway and GO cellular component enrichment analysis upon cold exposure (3 days). **c** Scatter plot of thermogenesis-related DEGs upon cold exposure (3 days). **d** DEGs interaction network in iWAT of HIF2α AKO mice upon cold exposure (3 days). **e** mRNA levels in iWAT from WT ($n = 7$), HIF1α AKO ($n = 6$), HIF2α AKO ($n = 6$), and HIF1/2α DKO ($n = 6$) mice upon cold exposure (3 days). **f, g** Western blot analysis of PKA Cα in iWAT from WT and HIF2α AKO mice upon **f** cold exposure (3 days) or **g** daily CL administration (0.5 mg/kg, 4 days). **h** Western blot analysis of PKA signaling and UCP1 in iWAT from WT and HIF2α AKO mice upon TN or cold exposure (6 h). **i** Western blot analysis of PKA signaling in iWAT from WT and HIF2α AKO mice upon CL administration (0.5 mg/kg, 4 h). **j** Western blot analysis of PKA signaling and UCP1 in beige adipocytes from WT and HIF2α AKO mice without or with ISO (5 μM, 1 h). **k** Glycerol concentration in culture media of beige adipocytes from WT ($n = 5$) and HIF2α AKO ($n = 5$) mice without or with ISO (5 μM, 3 h). **l** Intracellular cAMP levels of beige adipocytes from WT ($n = 4$) and HIF2α AKO ($n = 4$) mice without or with ISO (1 μM, 15 min). **m** mRNA levels in beige adipocytes from WT ($n = 3$) and HIF2α AKO ($n = 3$) mice. **n** Western blot analysis of PKA Cα in beige adipocytes from WT and HIF2α AKO mice. **o** mRNA levels in beige adipocytes infected with Admock ($n = 4$) or AdHIF2α ($n = 4$). **p** Western blot analysis of PKA Cα beige adipocytes infected with Admock or AdHIF2α. Data were expressed as the mean ± SEM by either two-tailed unpaired Student $t$-tests in (**m, o**), one-way ANOVA in (**e**), or two-way ANOVA in (**k, l**) followed by Holm–Sidak's multiple comparisons test. ISO isoproterenol.

determine the roles of HIF2α during re-warming, WT and HIF2α AKO mice were placed in a cold environment for 2 weeks, at which time the number of beige adipocytes appeared to reach a maximum level and to be comparable between the genotypes (Fig. 7b, c), and were then exposed to TN for 1 week. Intriguingly, beige-to-white transition upon re-warming was delayed in iWAT of HIF2α AKO mice (Fig. 7c). As miR-3085-3p could mediate the thermoregulatory function of HIF2α, the level of miR-3085-3p was examined. As shown in Fig. 7d, e, miR-3085-3p was down-regulated, concurrently with an increase of *Prkaca* upon re-warming. Although the expression of thermogenic genes in iWAT was attenuated upon re-warming (Fig. 7e), the levels of OXPHOS complexes, PKA Cα, and UCP1 in iWAT of HIF2α AKO mice were higher than those in WT mice (Fig. 7f, g), implying that increased PKA Cα in adipocyte HIF2α deletion might maintain mitochondrial abundance during re-warming.

Given that HIF2α-dependent mitochondrial regulation was mediated by miR-3085-3p in beige adipocytes (Fig. 6), mRNA levels of OXPHOS genes were examined. MiR-3085-3p mimic attenuated OXPHOS gene expression in HIF2α deficient beige adipocytes (Supplementary Fig. 9a), while miR-3085-3p inhibitor slightly but substantially rescued the expression of OXPHOS genes under HIF2α overexpression (Supplementary Fig. 9b). To assess the in vivo roles of miR-3085-3p in mitochondrial regulation, cold exposed mice were administered miR-3085-3p mimic at iWAT and subjected to re-warming (Fig. 7h). As indicated in Fig. 7i, the number of unilocular adipocytes in iWAT was increased by miR-3085-3p mimic. Further, the increase in OXPHOS complexes in iWAT of HIF2α AKO mice was suppressed by miR-3085-3p mimic, accompanied by down-regulation of PKA Cα and UCP1 (Fig. 7j). Meanwhile, overall autophagy-related genes and proteins were comparable between the genotypes (Supplementary Fig. 9c, d), implying that general autophagy might not be involved in the HIF2α-dependent whitening of beige adipocytes. Therefore, these data suggest that adipocyte HIF2α could induce beige-to-white transition via miR-3085-3p-dependent mitochondrial regulation during re-warming.

## Discussion

Thermogenesis has to be tightly fine-tuned to avoid the exhaustion of stored energy sources and cytotoxicity. Here, we demonstrate that cold-induced HIFα functions as a molecular brake on hyperactive thermogenesis in adipocytes. Mechanistically, HIF2α suppresses PKA Cα through miR-3085-3p expression, thereby mitigating thermogenic execution in beige adipocytes. These findings suggest that HIF2α would play a key regulatory role in the thermogenic programming of adipocytes via HIF2α-miR-3085-3p-PKA Cα axis (Fig. 8).

To maintain whole-body energy homeostasis, it is important to sense environmental alterations. Consistent with a previous

report[19], we found a cold-induced hypoxic environment in BAT and iWAT. HIFα protein expression was increased in the thermogenic adipose tissue upon cold or chronic CL treatment, probably, due to elevated UCP1-dependent oxygen consumption. In addition, recent studies have provided clues that HIFα could be induced in thermogenic adipose tissues independent of oxygen deprivation[19,36]. For example, increased mitochondrial ROS generation during OXPHOS could prevent PHD-dependent hydroxylation, leading to stabilization of HIFα in thermogenic adipocytes[37]. Also, accumulation of succinate in thermogenic adipose tissues might limit PHD activity upon cold[38]. Under various physiological and pathological conditions, including certain types of tumors, HIFα is activated by metabolic or micro-environmental alterations and relieves metabolic stress through modulating cellular pathways[39–41]. Thus, it is plausible to speculate that HIFα is a key sensor and responder for adaptation to the cellular metabolic status in thermogenic adipocytes.

While many studies have revealed various pro- and anti-thermogenic factors, their fine-tuning roles and underlying mechanisms are rather unclear. To prevent overheating which could be potentially triggered by hyperactive thermogenesis, the homeothermic function should be properly accomplished by sensing tissue-specific microenvironments and regulating cellular metabolic pathways to coordinate systemic homeostasis. In this study, we propose the roles of cold-induced HIF2α as a thermostat in adaptive thermogenesis. Several lines of evidence suggest that HIF2α would control thermogenic execution in beige adipocytes. First, HIF2α AKO mice exhibited cold-tolerant phenotypes with increased beige adipocytes upon cold exposure. Moreover, pharmacological modulation of HIFα and several adipocyte HIFα deficient mouse models including HIF1α AKO, HIF1/2α DKO, HIF1α BKO, and HIF2α BKO mice clearly showed that HIFα could suppress thermogenic function upon cold. Second, HIF2α attenuated PKA activity in beige adipocytes. Mechanistically, we elucidated that HIF2α suppressed PKA Cα expression via miR-3085-3p, resulting in downregulation of the PKA signaling cascade and thermogenic gene expression. Third, mitochondrial contents and activities were augmented in HIF2α deficient adipocytes, thereby increasing energy expenditure upon CL treatment. Conversely, HIF2α overexpression suppressed mitochondrial functions in beige adipocytes. Lastly, the retention of beige adipocytes was extended in HIF2α AKO iWAT during re-warming. In contrast, administration of miR-3085-3p promoted beige-to-white transition via PKA Cα suppression in iWAT of HIF2α AKO mice. Together, the present study proposes that HIF2α acts as a key safeguard to maintain physiologically appropriate thermogenesis.

Homeothermic animals control thermogenic activity during seasonal changes. In rodents and humans, the expression of thermogenic genes and beige marker genes is activated in adipose

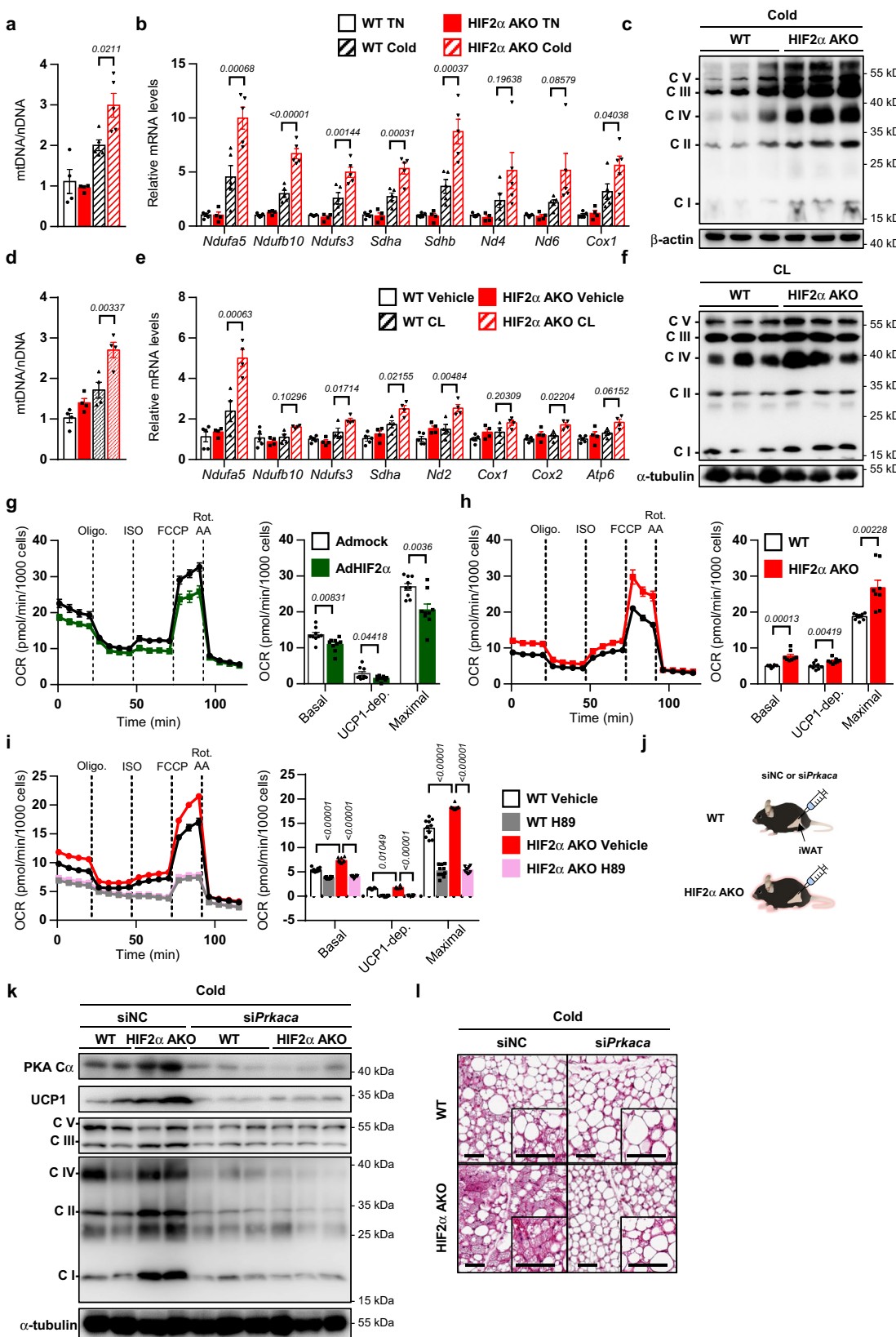

tissue during winter as compared to summer[42–44]. Including *Prkaca*, the expression levels of PKA-related genes are also elevated during hibernation in ground squirrels[45]. These findings raise the possibility that PKA subunit regulation might affect functional plasticity in beige adipocytes for adaptation to seasonal temperature changes. Here, we showed that PKA Cα would be a

critical node for the maintenance of beige characteristics as well as the generation of beige adipocytes. In particular, increased PKA Cα in HIF2α deficient beige adipocytes promoted PKA activity and thermogenic function. In contrast, PKA Cα suppression via miR-3085-3p boosted the whitening of beige adipocytes. As cAMP is a factor to mediate immediate feedback

**Fig. 5 HIF2α deficiency upregulates mitochondrial OXPHOS in beige adipocytes. a** Relative mtDNA in iWAT from WT (TN; $n = 4$, Cold; $n = 5$) and HIF2α AKO (TN; $n = 4$, Cold; $n = 5$) mice upon TN or cold exposure (3 days). **b** mRNA levels in iWAT from WT ($n = 5$) and HIF2α AKO (TN; $n = 4$, Cold; $n = 5$) mice upon TN or cold exposure (3 days). **c** Western blot analysis of OXPHOS complexes in iWAT from WT and HIF2α AKO mice upon cold exposure (3 days). **d** Relative mtDNA in iWAT from WT ($n = 4$) and HIF2α AKO ($n = 4$) mice upon daily CL administration (0.5 mg/kg, 4 days). **e** mRNA levels in iWAT from WT (Vehicle; $n = 5$, CL; $n = 4$) and HIF2α AKO ($n = 4$) mice upon daily CL administration (0.5 mg/kg, 4 days). **f** Western blot analysis of OXPHOS complexes in iWAT from WT and HIF2α AKO mice upon daily CL administration (0.5 mg/kg, 4 days). **g** OCRs and quantification in beige adipocytes infected with Admock ($n = 9$) or AdHIF2α ($n = 9$). **h** OCRs and quantification in beige adipocytes from WT ($n = 8$) and HIF2α AKO ($n = 8$) mice. **i** OCRs and quantification in beige adipocytes from WT ($n = 10$) and HIF2α AKO ($n = 10$) mice without or with 1 h preincubation of H89 (50 μM). **j** Experimental scheme of siPrkaca injection. **k** Western blot analysis of PKA Cα, UCP1, and OXPHOS complexes in iWAT from WT and HIF2α AKO mice with siNC or siPrkaca injection upon cold exposure (3 days). **l** Representative images of H&E staining of iWAT from WT and HIF2α AKO mice with siNC or siPrkaca injection upon cold exposure (3 days). Scale bars, 50 μm. Data were expressed as the mean ± SEM by two-tailed unpaired Student t-tests in (**g**, **h**) or two-way ANOVA in (**a**, **b**, **d**, **e**, **i**) followed by Holm–Sidak's multiple comparisons test. Oligo. oligomycin, ISO isoproterenol, Rot. rotenone, AA antimycin A.

loops, it appears that quantitative regulation of PKA subunits might be another effective way to sustain long-term thermogenic properties during chronic cold exposure. In this regard, it has been reported that chronic PKA activation via overexpression of Prkaca or deletion of Prkar2b increases UCP1 and contributes to the induction of beige adipocytes in iWAT, independent of cAMP changes[16,18]. Thus, it is feasible to propose that PKA subunit regulation might be important for the modulation of thermogenic programming in adipocytes upon metabolic and environmental stimuli.

Enhanced mitochondrial function is one of the key features of beige adipocytes. PKA signaling not only induces mitochondrial biogenesis but also stimulates mitochondrial respiration via phosphorylation of mitochondrial proteins[14,46]. Given that increased OXPHOS complexes and mitochondrial respiration in HIF2α deficient beige adipocytes were inhibited by miR-3085-3p, HIF2α-dependent PKA Cα suppression would be critical for mitochondrial regulation in beige adipocytes. Along with mitochondrial activation and biogenesis, PKA has been proposed to attenuate autophagy and mitophagy in beige adipocytes, leading to the preservation of mitochondrial contents[35,47]. Further, it has been reported that HIF1α induces mitophagy via Bnip3 and Bnip3l expression, and hypoxia activates Fundc1-mediated mitophagy[48,49]. However, the expression of general autophagy-related proteins and lysosomal genes did not differ between WT and HIF2α AKO iWAT upon re-warming. Although we cannot exclude the possibility that HIF2α might affect mitophagy and mitochondrial proteolytic pathways, our findings suggest that HIF2α would control the balance of mitochondrial functions, contributing to the regulation of beige adipocyte plasticity.

HIF1α and HIF2α share many target genes and participate in common cellular pathways. However, they also exert distinct roles and even opposing functions via their unique target gene expression. In this study, we found that HIF1α AKO and HIF2α AKO mice exhibited concordant phenotypes upon cold exposure. In accordance with our previous report[25], both HIF1α and HIF2α suppressed PKA activity in beige adipocytes. In line with this, HIF1α overexpression in adipose tissue suppressed whole-body energy expenditure and oxygen consumption in BAT[50]. Furthermore, it has been reported that HIF1α inhibitor PX-478 and HIF2α inhibitors PT2385 and PT2399 prevent diet-induced obesity and liver steatosis, accompanied by an induction of thermogenic gene expression in adipose tissues[51–53]. Especially, intestinal HIF2α could provoke insulin resistance via induction of ceramide salvage pathway and regulate thermogenesis via lactate-dependent remodeling of gut microbiome[52,54]. Here, we found that HIF2α seemed to have more potent effects than HIF1α in the regulation of thermogenic gene expression and energy expenditure. Notably, inhibition of PKA Cα by miR-3085-3p was selective to HIF2α, as no such regulation was observed upon HIF1α modulation. Although further studies for the detailed roles of

HIF1α and HIF2α remain to be elucidated, it is plausible to postulate that there may exist another HIF1α and/or HIF2α-dependent thermogenic regulation besides PKA Cα inhibition.

In conclusion, the present study proposes that HIF2α-dependent PKA regulation exerts a crucial role in the delicate control of thermogenesis in beige adipocytes. It is likely that HIF2α coordinates the plasticity of beige adipocytes in order to strengthen adaptation to a temperature shift as well as fine-tune thermogenesis, to eventually avoid futile energy wastage. Given that thermogenic adipocytes play pivotal roles in whole-body energy expenditure, it seems that these findings may provide clues for the development of promising interventions to counteract obesity and related complications by promoting energy consumption.

## Methods

**Animals.** All experiments with mice were approved by the Seoul National University Institutional Animal Care and Use Committee. HIF1α AKO, HIF2α AKO, and HIF1/2α DKO mice were generated by crossing Adiponectin-Cre mice with Hif1a$^{flox/flox}$, Hif2a$^{flox/flox}$, and Hif1a$^{flox/flox}$/Hif2a$^{flox/flox}$ mice (C57BL/6-Hif1a$^{tm3Rsjo}$/J and C57BL/6-Hif2α$^{tm1Mcs}$/J). HIF1α BKO and HIF2α BKO mice were generated by crossing Ucp1-Cre mice with Hif1a$^{flox/flox}$ and Hif2a$^{flox/flox}$ mice. Mice were housed in groups of 3–5 mice per cage with 22 °C and 55% relative humidity conditions and maintained under 12-h/12-h light/dark cycles with free access to water and a normal chow diet (Ziegler Feed; DBL; 22.4% protein, 4.88% lipid of total calories). For TN and cold exposure experiments, 10–12-week-old male mice (1–2 mice per cage) were placed in an environmental cabinet (Environmental Cabinet, DBL Co.) at 30 °C or 4–6 °C, respectively. Rectal temperature was measured using a thermal probe (Testo925, Testo Inc.). Infrared thermography was conducted using an infrared camera (CX320, COX Co.). For in vivo HIFα modulation, male mice received daily intraperitoneal injections of YC-1 (30 mg/kg) or DMOG (40 mg/kg) from 1 day before cold exposure to 3 days after cold exposure. For pharmacological inhibition of HIF2α, male mice were given PT2385 (10 mg/kg) orally twice a day from 1 day before cold exposure to 3 days after cold exposure. PT2385 was dissolved in a mixture of 10% Ethanol, 40% PEG300, 5% Tween-80, and 45% saline. For beige adipocytes induction via β3-adrenergic signaling, CL-316,243 (0.5 mg/kg, C5976, Sigma) was intraperitoneally injected into male mice for 4 consecutive days. For siRNA or miRNA mimic delivery, 4 μg of siPrkaca, miR-3085-3p, or control mimic was directly injected into iWAT of male mice using in vivo-jetPEI® (201-10 G, Polyplus) according to the manufacturer's protocol.

**Cell culture and transfection.** Stromal vascular fraction (SVF) was isolated from iWAT. For beige adipogenesis, SVF was grown to confluence in Dulbecco's modified Eagle's medium (DMEM) supplemented with 10% fetal bovine serum (FBS), 100 units/mL penicillin, and 100 μg/mL streptomycin. To induce beige adipocyte differentiation, cells were incubated with DMEM containing 10% FBS, 2 μg/ml dexamethasone (Dex), 0.5 mM 3-isobutyl-1-methylxanthine (IBMX), 2 μM rosiglitazone, 125 nM indomethacin, 1 nM 3,3,5-triiodo-L-thyronine (T3), and 167 nM insulin for 2 days. Then, the culture medium was replaced with DMEM containing 10% FBS, 2 μM rosiglitazone, 1 nM T3, and 167 nM insulin and the cells were cultured for additional 2 days. The culture medium was changed with DMEM containing 10% FBS every other day. For β-adrenergic activation, cells were treated with 5 μM ISO. Immortalized murine brown adipocytes (BACs) were grown to confluence in DMEM supplemented with 10% FBS, 100 units/mL penicillin, and 100 μg/mL streptomycin and incubated in induction medium consisting of DMEM, 10% FBS, 20 nM insulin, 1 nM T3, 125 nM indomethacin, 0.5 mM IBMX, and 2 μg/mL Dex for 2 days. The culture medium was then replaced with DMEM containing 10% FBS, 20 nM insulin, and 1 nM T3 for 2 additional days. The culture medium

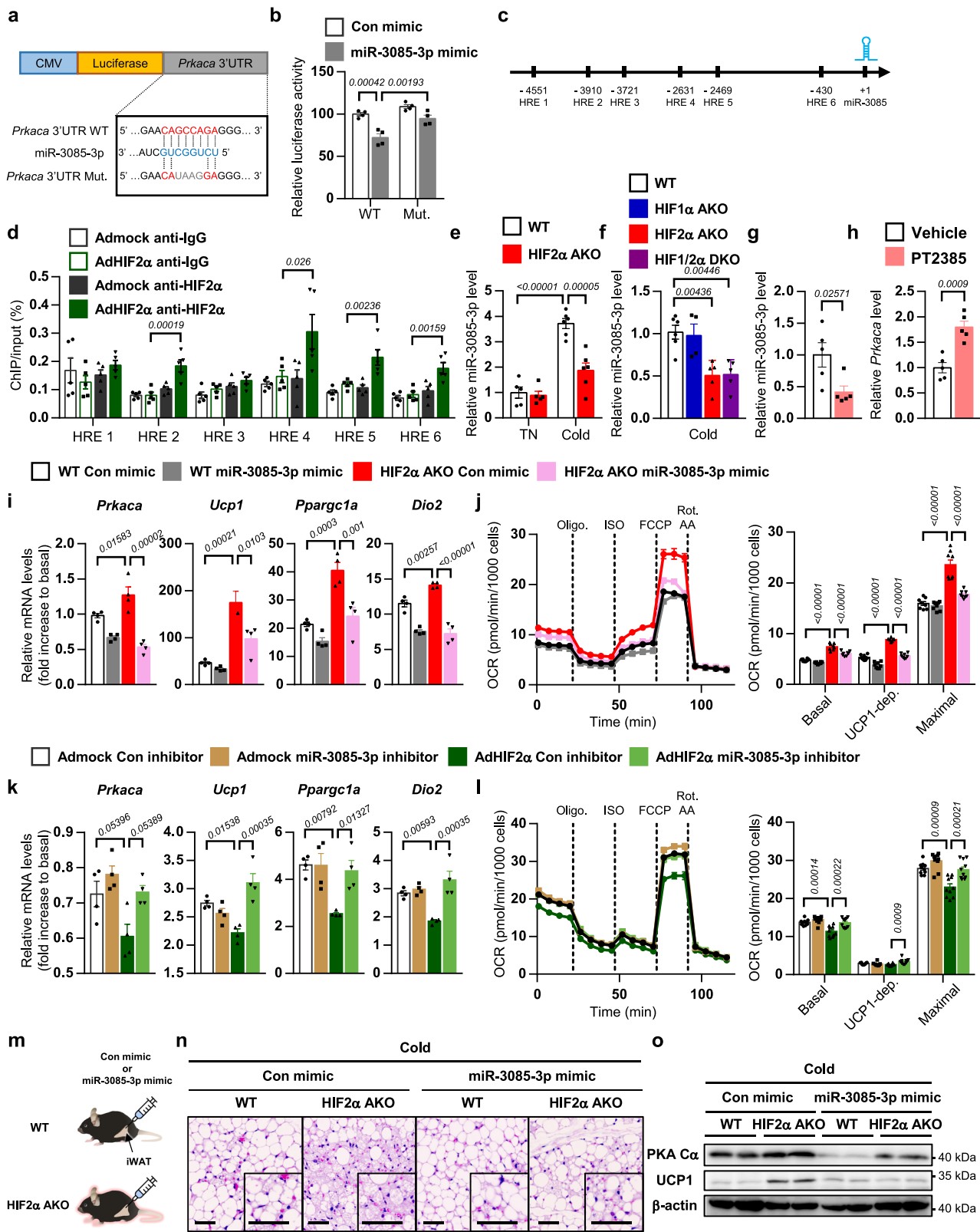

was changed every other day with DMEM containing 10% FBS. HEK293FT cells were grown in DMEM supplemented with 10% FBS, 100 units/mL penicillin, and 100 μg/mL streptomycin.

**RNA-sequencing analysis**. Raw sequence reads were trimmed and quality-controlled using TrimGalore (version 0.4.4). The trimmed reads were mapped to the mouse genome GRCm38/mm10 using the STAR aligner (version 2.6.1d)[55]. Raw read counts per gene were computed using HTSeq-count (version 0.13.5)[56].

Significant DEGs were identified using DESeq2 (version 1.30.0)[57]. The significance thresholds were |log2FC| > 0.2 and $P < 0.15$ for cold HIF2α AKO samples versus cold WT samples. Gene ontology enrichment analysis of the upregulated DEGs in cold HIF2α AKO samples was conducted using EnrichR[58].

**Network analysis**. Network analysis was performed to prioritize candidate genes at the network level that would mediate thermogenesis and mitochondrial biogenesis in HIF2α AKO iWAT. A *Mus musculus* protein-protein interaction (PPI)

**Fig. 6 miR-3085-3p mediates HIF2α-dependent PKA Cα regulation in beige adipocytes. a** *Prkaca* 3'UTR sequence and predicted miR-3085-3p binding site. **b** Luciferase reporter activities of WT ($n = 4$) and mutant ($n = 4$) *Prkaca* 3'UTR upon con or miR-3085-3p transfection. **c** Upstream region of miR-3085-3p and hypoxia-response element (HRE) sites. **d** ChIP-qPCR analysis of BAC cells upon Admock ($n = 5$) or AdHIF2α ($n = 5$) infection. **e** miR-3085-3p level in iWAT from WT (TN; $n = 5$, Cold; $n = 6$) and HIF2α AKO (TN; $n = 5$, Cold; $n = 6$) mice upon TN or cold exposure (3 days). **f** miR-3085-3p level of iWAT from WT ($n = 6$), HIF1α AKO ($n = 4$), HIF2α AKO ($n = 5$), and HIF1/2α DKO ($n = 5$) mice upon cold exposure (3 days). **g** miR-3085-3p and **h** *Prkaca* levels in iWAT from vehicle- ($n = 5$) and PT2385- ($n = 5$) administered mice upon cold exposure (3 days). **i** mRNA levels in beige adipocytes from WT ($n = 4$) and HIF2α AKO ($n = 4$) mice upon con or miR-3085-3p mimic transfection with ISO (5 μM, 4 h). **j** OCRs and quantification in beige adipocytes from WT ($n = 8$) and HIF2α AKO ($n = 8$) mice upon con or miR-3085-3p mimic transfection. **k** mRNA levels in beige adipocytes infected with Admock ($n = 4$) or AdHIF2α ($n = 4$) upon con or miR-3085-3p inhibitor transfection with ISO (5 μM, 4 h). **l** OCRs and quantification in beige adipocytes infected with Admock ($n = 10$) or AdHIF2α ($n = 10$) upon con or miR-3085-3p inhibitor transfection. **m** Experimental scheme of miRNA mimic injection. **n** Representative images of H&E staining of iWAT from WT and HIF2α AKO mice with con or miR-3085-3p mimic injection upon cold exposure (3 days). Scale bars, 50 μm. **o** Western blot analysis of PKA Cα and UCP1 in iWAT from WT and HIF2α AKO mice with con or miR-3085-3p mimic injection upon cold exposure (3 days). Data were expressed as the mean ± SEM by two-tailed unpaired Student *t*-tests in (**g**, **h**), one-way ANOVA in (**f**), or two-way ANOVA in (**b**, **d**, **e**, **i–l**) followed by Holm–Sidak's multiple comparisons test. Oligo. oligomycin, ISO isoproterenol, Rot. rotenone, AA antimycin A.

network from the STRING database (version 10.5) was used to construct a condition-specific DEG interaction network[59]. Network nodes consisted of *Hif2a* with DEGs between WT and HIF2α AKO iWAT upon cold exposure, and network edges consisted of high-confidence STRING PPI edges (confidence score >0.55). Network propagation was used to rank genes based on their relevance to *Hif2a* in the network by the Walker module using *Hif2a* as a seed gene[60]. For centrality analysis, betweenness centrality was used to rank genes based on centrality in the network topology using the NetworkX package (version 2.2). To analyze thermogenic genes, Wikipathway gene annotation was used[61], and gene ontology annotation related to thermogenic genes was assessed using mouse genome informatics[62]. Network nodes were clustered using the Markov Cluster Algorithm in clusterMaker2 (version 1.3.1) in Cytoscape. The network was visualized using the Cytoscape software.

**Indirect calorimetry.** Indirect calorimetry was performed using PhenoMaster (TSE Systems) according to the manufacturer's protocol. Twelve-week-old male mice were placed in a calorimetric chamber for 48 h prior to measurements of VO$_2$, VCO$_2$, and energy expenditure. To activate β3-adrenergic signaling, mice were intraperitoneally injected with CL-316,243 (0.5 mg/kg). The experiments were performed at the Korea Mouse Phenotyping Center (KMPC), SNU, Seoul, Korea.

**Tissue histology and fluorescence imaging.** Adipose tissue was isolated from mice, fixed in 4% paraformaldehyde, and embedded in a paraffin block. Paraffin blocks were cut into 5 μm sections at 30 μm intervals. The sections were stained with H&E following the standard protocol and imaged using a digital slide scanner (Axio Scan Z1, Carl Zeiss). For hypoxia staining, male mice were intraperitoneally injected with pimonidazole HCl (50 mg/kg, Hypoxyprobe) 1 h before sacrifice, and their adipose tissues were embedded in paraffin. After deparaffinization and rehydration, tissue slides were stained with mouse-anti-pimonidazole (1:50, MAb1, Hypoxyprobe) and visualized by DAB staining (SK-4105, Vector Laboratories). For immunostaining, deparaffinized slides were stained with anti-PLIN1 (1:400, 20R-PP004, Fitzgerald), anti-HIF1α 1:200, NB100-449, Novus, anti-HIF2α 1:200, NB100-122, Novus, and anti-UCP1 (1:400, ab10983, Abcam) antibodies in blocking solution containing 10% horse serum at 4 °C overnight. Then, the sections were washed with PBS and stained with species-specific secondary antibodies and Hoechst (1:1000, H3570, Thermo Fisher Scientific) for 1 h. After applying the mounting solution (VECTASHIELD without DAPI, Vector Laboratories), the specimens were observed using a coherent anti-Strokes Raman scattering (CARS) microscope (TCS SP8 CARS microscope, Leica Microsystems). For JC-1 staining, beige adipocytes were incubated with JC-1 (5 μg/mL, T3168, Thermo Fisher Scientific) at 37 °C for 30 min. After washing, the cells were imaged using a CARS microscope.

**cAMP measurement.** cAMP concentrations in beige adipocytes were measured using a direct cAMP ELISA kit (Cat. no. 25-0114, Enzo Life Sciences) according to the manufacturer's protocol. Beige adipocytes were lysed in 0.1 M HCl with 0.1% Triton X-100. The cell lysates were subjected to ELISA. To induce cAMP production, cells were treated with 1 μM ISO (I6504, Sigma) for 15 min. The results were analyzed using four parametric logistic curve fitting models. Total protein contents were measured to normalize the cAMP concentrations.

**Immunoblotting.** Adipose tissues and cells were lysed on ice with modified RIPA buffer containing 50 mM Tris-HCl (pH 7.5), 150 mM NaCl, 2 mM EDTA, 1% Triton X-100, 0.5% sodium deoxycholate, 0.1% sodium dodecyl sulfate (SDS), 5 mM NaF, 1 mM Na$_3$VO$_4$, and a protease inhibitor cocktail (#P3100, GeneDEPOT). Protein lysates were boiled and subjected to SDS-polyacrylamide gel electrophoresis. The separated proteins were transferred onto polyvinylidene fluoride membranes. Antibodies against HIF1α (NB100-479, Novus), HIF2α (NB100-132, Novus), UCP1 (ab10983, Abcam), PKA Cα (4782S, Cell Signaling), HSL (4107S,

Cell Signaling), pHSL (4139S, Cell Signaling), CREB (9197S, Cell Signaling), pCREB (9191S, Cell Signaling) pPKA substrate (9624S, Cell Signaling), pP38 (612289, BD Biosciences), P38 (Sc-7972, Santa Cruz Technology), OXPHOS (ab110413, Abcam), α-tubulin (T6199, Sigma), β-actin (A5441, Sigma), and Lamin B1 (ab16048, Abcam) were used. Protein bands were visualized with horseradish peroxidase-conjugated secondary anti-rabbit IgG (A0545, Sigma), and anti-mouse IgG (A9044, Sigma).

**Glycerol release assay.** Glycerol levels were measured using Free Glycerol Reagent (F6428, Sigma) according to the manufacturer's protocol. For ex vivo lipolysis, isolated iWAT was cut into 30-mg pieces and incubated at 37 °C in DMEM containing 3% FA-free BSA in the absence or presence of 5 μM ISO for 3 h. For in vitro lipolysis, differentiated beige adipocytes were incubated at 37 °C in DMEM containing 3% FA-free BSA in the absence or presence of 5 μM ISO for 3 h.

**Quantification of mitochondrial DNA.** DNA was isolated from iWAT or beige adipocytes using phenol-chloroform extraction. mtDNA abundance was measured by qPCR using mitochondrial genomic primers for NADH ubiquinone oxidoreductase chain 4 (ND4). The nuclear 18 S rRNA gene was used for normalization. The sequence of the primer used for qPCR are listed in Supplementary Table 1.

**Cellular oxygen consumption assay.** The cellular OCR was analyzed using a Seahorse XFe96 extracellular flux analyzer (Agilent) according to the manufacturer's instructions. Differentiated beige adipocytes were incubated in assay medium (25 mM glucose, 1 mM sodium pyruvate, 2 mM L-glutamine, and 1% fatty acid-free BSA in Seahorse XF base medium at pH 7.4). To evaluate mitochondrial activity, the OCR was measured following treatment with 5 μM oligomycin (75371, Sigma), 1 μM ISO, 5 μM carbonyl cyanide-p-trifluoromethoxyphenylhydrazone (FCCP, C2920, Sigma), 5 μM antimycin A (A8674, Sigma), and 5 μM rotenone (R8875, Sigma). For PKA inhibition, cells were pre-treated with 50 μM H89 (B1427, Sigma) for 1 h. Cell numbers were determined through Hoechst staining and used to normalize the OCRs. Parameter values were calculated using previously reported equations[63]. Basal respiration: value prior to the injection of oligomycin minus non-mitochondrial respiration. UCP1-dependent respiration: maximal value after the injection of ISO minus minimal value after injection of oligomycin. Maximal respiration: maximal value after the injection of FCCP minus non-mitochondrial respiration. Non-mitochondrial respiration: minimal value after the injection of rotenone and antimycin A.

**Chromatin immunoprecipitation qPCR.** BACs were crosslinked with 1% formaldehyde for 20 min and lysed with lysis buffer (1% SDS, 10 mM EDTA, 50 mM Tris-HCl [pH 8.1], and protease inhibitor cocktail). The samples were diluted with a dilution buffer (0.01% SDS, 1.1% Triton X-100, 1.2 mM EDTA, 16.7 mM Tris-HCl [pH 8.1], 167 mM NaCl, and protease inhibitor cocktail) and sonicated for 15 min. After being precleared with protein A agarose (17-0780-01, GE Healthcare) and salmon sperm DNA, the samples were immunoprecipitated with antibodies overnight. The immunoprecipitated samples were collected by adding protein A-sepharose beads and sequentially washed with low salt buffer (0.1% SDS, 1% Triton X-100, 2 mM EDTA, 20 mM Tris-HCl [pH 8.1], and 150 mM NaCl), high salt buffer (0.1% SDS, 1% Triton X-100, 2 mM EDTA, 20 mM Tris-HCl [pH 8.1], and 500 mM NaCl), LiCl buffer (0.25 M LiCl, 1% NP-40, 1 mM EDTA, 10 mM Tris-HCl [pH 8.1]) three times, and TE buffer (10 mM Tris-HCl [pH 8.1] and 1 mM EDTA) twice. Then, the samples were eluted with elution buffer (1% SDS and 0.1 M NaHCO$_3$), and protein–DNA cross-linking was reversed by incubation with 200 mM NaCl at 65 °C for 12 h. After reverse cross-linking, DNA was extracted using phenol-chloroform extraction. Precipitated DNA fragments were analyzed by qPCR. The sequence of the primer used for ChIP-qPCR are listed in Supplementary Table 1. Antibodies against HIF2α (NB100-132, Novus) and mouse IgG (A9044, Sigma) were used for immunoprecipitation.

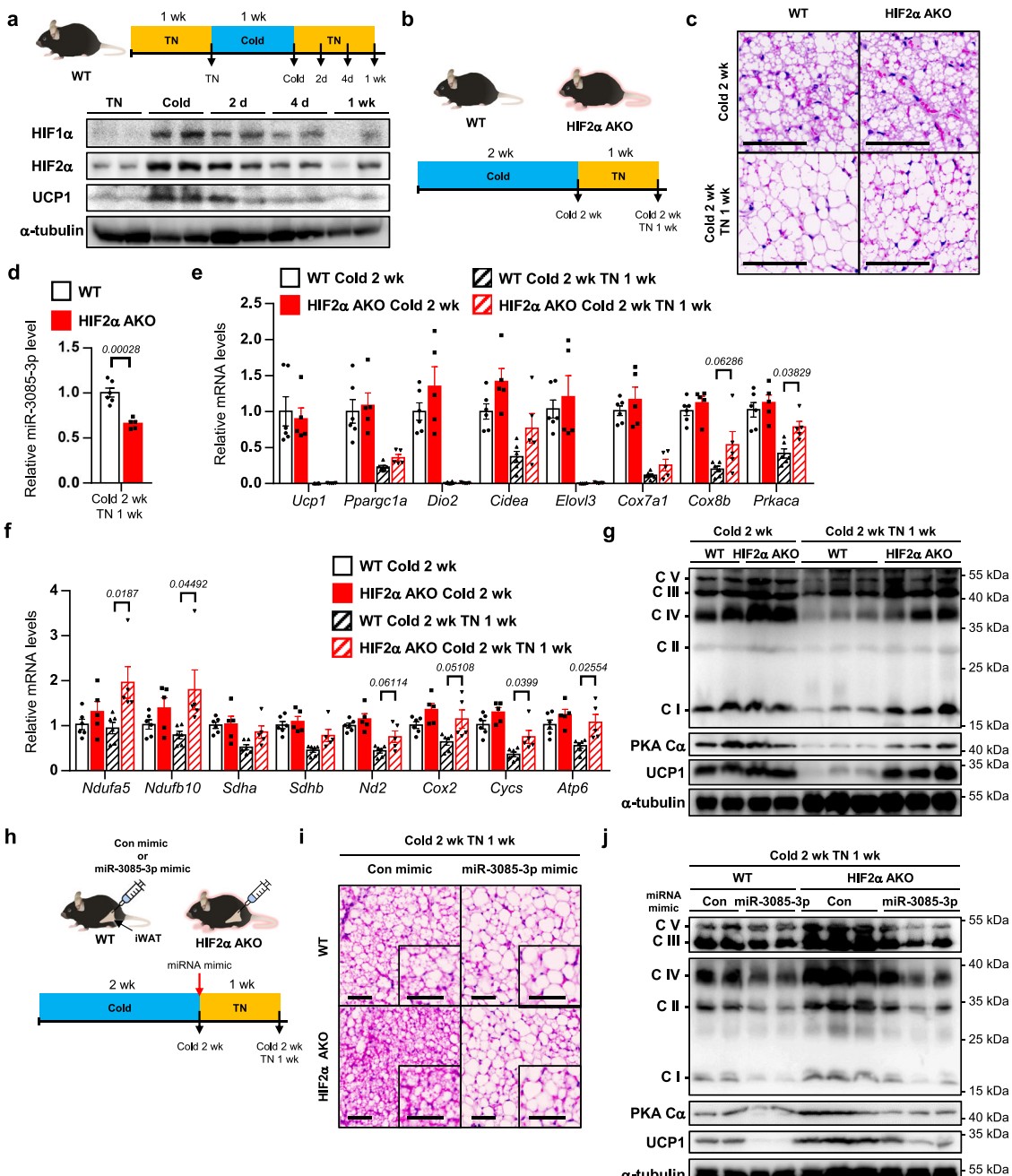

**Fig. 7 During the re-warming process, adipocyte HIF2α deficiency fails to mitigate the mitochondrial activity. a** Experimental scheme (top) and western blot analysis of HIFα and UCP1 in iWAT of upon cold exposure or re-warming (bottom). **b** Experimental scheme of re-warming. **c** Representative images of H&E staining of iWAT from WT and HIF2α AKO mice upon cold exposure or re-warming (cold 2 weeks + TN 1 week). Scale bars, 100 μm. **d** miR-3085-3p level of iWAT from WT (n = 6) and HIF2α AKO (n = 5) mice upon cold and re-warming (cold 2 week + TN 1 week). **e, f** mRNA levels in iWAT from WT (n = 6) and HIF2α AKO (n = 5) mice upon cold exposure and re-warming (cold 2 weeks + TN 1 week). **g** Western blot analysis of OXPHOS complexes, PKA Cα, and UCP1 in iWAT from WT and HIF2α AKO mice upon cold exposure and re-warming (cold 2 weeks + TN 1 week). **h** Experimental scheme of re-warming with con or miR-3085-3p mimic injection. **i** Representative images of H&E staining of iWAT from WT and HIF2α AKO mice upon re-warming (cold 2 weeks + TN 1 week) with con or miR-3085-3p mimic injection. Scale bars, 50 μm. **j** Western blot analysis of OXPHOS complexes, PKA Cα, and UCP1 in iWAT from WT and HIF2α AKO mice upon re-warming (cold 2 weeks + TN 1 week) with con or miR-3085-3p mimic injection. Data were expressed as the mean ± SEM by two-tailed unpaired Student t-tests in (**d**) or two-way ANOVA in (**e, f**) followed by Holm–Sidak's multiple comparisons test.

**Adenovirus infection**. HIF1α and HIF2α adenoviruses were generously provided by Dr. Jang-Soo Chun (Gwangju Institute of Science and Technology, Gwangju, South Korea). Green fluorescent protein-containing adenovirus was used as a negative control (mock). Four days after differentiation, beige adipocytes or BACs were incubated with DMEM containing 10% FBS and adenovirus (at a multiplicity of infection of 500) for 16 h. The culture medium was replaced with fresh medium for 2 days.

**Prkaca 3′UTR target miRNA prediction and validation**. TargetScan mouse 7.2[64] and miRDB Version 6.0[65] were used to predict miRNA candidates targeting Prkaca 3′UTR. For luciferase assays, pGL3UC-Prkaca 3′UTR WT or pGL3UC-Prkaca 3′UTR Mut. construct was transfected with control or miR-3085-3p mimic into HEK293FT cells using Lipofectamine 3000 (L3000001, Thermo Fisher Scientific). After 24 h of transfection, cell lysates were analyzed for luciferase activity. A pCMV-β-galactosidase plasmid was used as an internal control for transfection

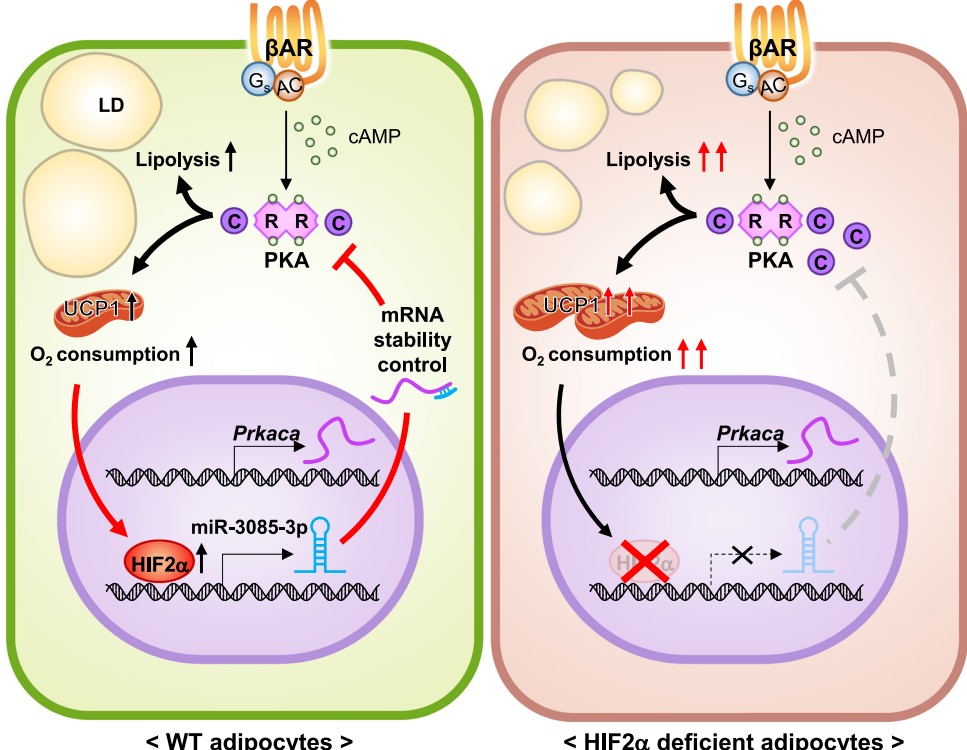

< WT adipocytes >          < HIF2α deficient adipocytes >

**Fig. 8 Proposed model.** Upon cold exposure, an activated thermogenic program in beige adipocytes stimulates oxygen consumption, thereby resulting in the stabilization of HIFα. Among the HIFα isoforms, upregulated HIF2α fine-tunes thermogenic execution via miR-3085-3p-dependent *Prkaca* regulation. However, adipocyte HIF2α deficiency augments PKA signaling and potentiates thermogenic functions, leading to the retention of beige adipocytes. Our findings suggest that the HIF2α-miR-3085-3p-PKA Cα axis forms negative feedback for appropriate regulation of thermogenesis.

efficiency. Primer information for the cloning and mutagenesis of Prkaca 3′UTR is listed in Supplementary Table 1.

**miRNA and siRNA transfection.** The mimic and inhibitor of miR-3085-3p and si*Prkaca* were purchased from Bioneer Inc. (Daejeon, Korea). For transfection, beige adipocytes were transfected with 100 nM siRNA, miRNA mimic, or inhibitor using Lipofectamine 3000 according to the manufacturer's protocol after 4 days of differentiation. For combined adenovirus infection and miRNA inhibitor transfection, adenoviral infection was performed prior to miRNA inhibitor transfection. After 16 h of infection with adenovirus, the culture medium was replaced with a fresh medium containing the miRNA inhibitor with Lipofectamine 3000. Sequence information for the siRNA and miRNAs is listed in Supplementary Table 1.

**RT-qPCR.** Total RNA was isolated from tissues or cells using TRIzol Reagent (RiboEx, GeneAll) and subjected to cDNA synthesis using the ReverTra Ace qPCR RT Kit (Toyobo). Relative mRNA levels were detected using the CFX96™ Real-Time System (Bio-Rad Laboratories). RT-qPCRs were run using SYBR Green Master Mix (DQ384-40h, Biofact). Target gene expression levels were normalized to cyclophilin gene expression levels. The primers used for RT-qPCR are listed in Supplementary Table 1. For miRNA extraction, Direct-zol™ RNA MiniPrep (Zymo Research) was used. The miRNA was reverse-transcribed using the MicroRNA Reverse Transcription Kit (4366596, Thermo Fisher Scientific) and TaqMan MicroRNA Assay (4440886, 4427975, Thermo Fisher Scientific). RT-qPCRs were run using TaqMan Master Mix (RT600S, Enzynomics). The miR-3085-3p level was normalized to that of snoRNA202.

**Statistics and reproducibility.** Data were presented as the mean ± standard error of the mean (SEM). In the figures, sample numbers and sizes are indicated by dots. In immunoblotting and DNA gel blotting, representative result from three independent experiments is shown. Representative images of H&E staining and immunohistochemistry were obtained from 4–5 replicates in each group. Comparisons between two groups were performed using a two-tailed unpaired Student $t$-test. Multiple comparisons were performed using one-way analysis of variance (ANOVA) or two-way ANOVA when two conditions were involved. Statistical analyses were conducted using GraphPad Prism 7, $p < 0.05$ was considered significant.

**Reporting summary**. Further information on research design is available in the Nature Research Reporting Summary linked to this article.

## Data availability
The accession number for RNA-seq data in this study is GSE179385. Source data are provided with this paper.

## Code availability
Custom scripts for network analysis is available on Github [https://github.com/minsik-bioinfo/HIF2a_PKA_NetworkAnalysis].

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

## Acknowledgements

This study was supported by the National Research Foundation, funded by the Korean government (NRF-2020R1A3B2078617). J.S.H., Y.G.J., G.L., H.N., S.M.H., J.C., Y.Y.K., K.C.S., and J.K were supported by the BK21 Plus program.

## Author contributions

J.S.H. and J.B.K. conceived and designed the project. J.S.H. performed most of the experiments, analyzed data, and wrote the manuscript. Y.G.J., M.O., and K.J. analyzed RNA sequencing and performed network propagation and centrality analysis. Y.G.J., G.L., H.N., S.M.H., J.C., Y.Y.K., K.C.S., and J.K contributed to performing in vivo

experiments and discussed the data. S.S.C., E.J.P., and S.K. contributed to the design experiments and discussed the data. J.B.K. supervised the study and wrote the manuscript.

## Competing interests
The authors declare no competing interests.
