## [Peer Review File · Nature Communications]

Adipocyte HIF2 α functions as a thermostat via PKA C α regulation in beige adipocytesREVIEWER COMMENTS

Reviewer #1 (Remarks to the Author):

The manuscript by Han et al. reports that cold and b-adrenergic receptor signaling induces HIFa signaling that acts as a molecular brake on thermogenesis by downregulating PKA. First, the authors generated fat-specific HIF1a, HIF2a, and HIF1/2a KO mice and demonstrated that an inhibition of HIFa signaling potentiated thermogenesis in beige fat. Next, the authors showed that HIF2a loss in adipocytes increases whole-body energy expenditure upon the activation of b-AR. Subsequently, through transcriptional analyses of the WAT in KO mice, the authors found that HIF2a loss was accompanied by upregulation of PKA-Ca and mitochondrial activity. Intriguingly, the authors identified miR-3085-3p through which HIF2a loss upregulates PKA-Ca. This regulatory pathway is involved in the beige-to-white adipocyte conversion as fat-specific HIF2a deletion leads to the retention of beige fat, whereas miR-3085 mimic promotes the whitening of beige fat in vivo.

Together, the study provided compelling evidence that cold-induced HIF2a is a crucial negative regulator of PKA signaling and beige fat biogenesis/retention. The research group is a front runner in the field of adipose tissue biology and gained a strong reputation for rigorous research. Overall, the data are compelling, and the manuscript is well-written. The reviewer suggests a few minor points that the authors want to address prior to publication.

1. HIF1a pathway is known to induce adipose tissue fibrosis, whereas recent studies suggest that beige fat biogenesis is accompanied by repressed fibrosis. It would be insightful if the authors discuss or look into the regulation of adipose tissue fibrosis by the HIF2a-miR-3085 axis.
2. The authors wish to mention if the role of miR-3085 is selective to beige adipocytes in the inguinal WAT or also in iBAT.

Reviewer #2 (Remarks to the Author):

In this manuscript Ji Seul Han et al. show that hypoxia-inducible factors -1 and -2 (HIF1 and HIF2) - are induced in inguinal WAT (iWAT) and brown adipose tissue (BAT) upon cold exposure. This compromises the expression of certain genes involved in thermogenesis and therefore loss of HIF1a and/or HIF2a in adipocytes (or BAT-specifically) augments surface body and rectal temperatures. Authors proposed a major role of HIF2 in this anti-thermogenic program. Indeed HIF2a induces miR-3085-3p, which in turns

represses PKAC α and therefore a protein kinase A (PKA)-dependent thermogenic/mitochondrial program in adipose tissue. Data are interesting but authors should address the following points.

Major comments:

1.- Authors show that mice treated with the HIF inhibitor YC-1 increased rectal temperature concomitant with an increased expression of thermogenic genes of iWAT upon cold exposure. YC-1 has the potential to inhibit both HIF1 and HIF2 expression (PMID: 19074848). Therefore authors should use the HIF2 specific inhibitors such as PT2385 or PT2399 (PMID: 27595394; PMID: 27595393; PMID: 31999648) to assess (i) whether this inhibitor prevents HIF2 protein accumulation in iWAT and BAT of cold-exposed mice and (ii) whether it increases rectal temperature and surface body temperature, iWAT expression of thermogenic genes as well as miR-3085-3p expression. This suggested experiment might broaden the knowledge about potential side effects of this HIF2a inhibitor primarily oriented to treat Vhl-deficient renal cell carcinoma. Moreover authors should show whether YC-1 treatment inhibits iWAT and BAT HIF1a and HIF2a protein expression and conversely whether DMOG induces the expression of these two isoforms.

2.- Authors claim that the HIF2 α -miR-3085-3p axis is central to reduce the expression of thermogenic genes as well as mitochondrial activity through the repression of PKAC α . This conclusion is drawn based on the use of a PKA inhibitor H89. Authors should use another approach with - for example - shRNA or siRNA against PKAC α in HIF2 α -deficient and control beige adipocytes and - if possible - after in vivo local administration in iWAT as authors do with miR-3085-3p mimic.

3.- Authors claim that HIF1a AKO mice show a thermogenic phenotype independently of miR-3085-3p. However authors should assess whether HIF1a AKO mice do not have elevated PKAC α expression but still can show altered mitochondrial complexes expression in iWAT after cold exposure.

4.- HIF1a and HIF2 deficient mice in brown adipocytes (HIF1BKO and HIF2a BKO) show increased body temperature but without major changes in the RNA expression of the thermogenic genes analyzed especially in HIF1a BKO. These data can lead to consider that metabolic changes described in iWAT might not be central to explain the increased mouse temperature? Or alternatively, do these HIF1BKO and HIF2a BKO mice have increased expression of thermogenic genes and PKAC α in iWAT possibly reflecting a possible interplay between BAT and iWAT?

5.- In page 19 (Discussion section) authors mention "Furthermore, it has been reported that HIF1 α inhibitor PX-478 and HIF2 α inhibitors PT2385 and PT2399 prevent diet-induced obesity and induce thermogenic gene expression in adipose tissues 50, 51, 52". Authors should provide more details about the role of HIF2a in thermogenesis shown in these references since in one of them (PMID: 29035368) its

role in related to intestinal HIF2a activity and not locally in the adipose tissue as shown in this manuscript. Moreover authors should cite and discuss the following two references related to the role of HIF signaling in thermogenesis (PMID: 27738746; PMID: 34329568) especially the one in which thermogenesis is impaired upon HIF1 α expression in brown adipose tissue (PMID: 27738746).

Minor comment:

1.- In page 7 (Results section) authors mention “Simultaneously, the levels of HIF1 α and HIF2 α proteins gradually increased during cold exposure in iWAT and BAT (Fig. 1c, d), but not in eWAT (Supplementary fig. 1b)”. However a modest HIF2 α induction was detected in eWAT upon cold exposure. Therefore authors should change this sentence accordingly.

Response to the reviewers' comments

manuscript NCOMMS-21-32870

Adipocyte HIF2 α functions as a thermostat via PKA C α regulation in beige adipocytes

Reviewer #1 (Remarks to the Author):

The manuscript by Han et al. reports that cold and β -adrenergic receptor signaling induces HIF α signaling that acts as a molecular brake on thermogenesis by downregulating PKA. First, the authors generated fat-specific HIF1 α , HIF2 α , and HIF1/2 α KO mice and demonstrated that an inhibition of HIF α signaling potentiated thermogenesis in beige fat. Next, the authors showed that HIF2 α loss in adipocytes increases whole-body energy expenditure upon the activation of β -AR. Subsequently, through transcriptional analyses of the WAT in KO mice, the authors found that HIF2 α loss was accompanied by upregulation of PKA-C α and mitochondrial activity. Intriguingly, the authors identified miR-3085-3p through which HIF2 α loss upregulates PKA-C α . This regulatory pathway is involved in the beige-to-white adipocyte conversion as fat-specific HIF2 α deletion leads to the retention of beige fat, whereas miR-3085 mimic promotes the whitening of beige fat in vivo.

Together, the study provided compelling evidence that cold-induced HIF2 α is a crucial negative regulator of PKA signaling and beige fat biogenesis/retention. The research group is a front runner in the field of adipose tissue biology and gained a strong reputation for rigorous research. Overall, the data are compelling, and the manuscript is well-written. The reviewer suggests a few minor points that the authors want to address prior to publication.

1. HIF1 α pathway is known to induce adipose tissue fibrosis, whereas recent studies suggest that beige fat biogenesis is accompanied by repressed fibrosis. It would be insightful if the authors discuss or look into the regulation of adipose tissue fibrosis by the HIF2 α -miR-3085 axis.

Thanks for raising the interesting viewpoint. As the reviewer pointed out, adipose tissue fibrosis is negatively correlated with beige adipogenesis. It has been reported that adipose tissue fibrosis could be repressed by the stimuli promoting beige adipocytes including cold exposure and chronic CL-316,243 administration^{1, 2, 3}. Especially, PRDM16 plays an important role in adipose tissue fibrosis via EHMT1-GTF2IRD1 complex-mediated suppression of fibrosis-related gene expression¹. In contrast, TGF β signaling, one of the crucial factors promoting fibrosis, attenuates beige adipogenesis and thermogenic programming^{4, 5, 6}. In obesity, adipocyte HIF1 α stimulates adipose tissue fibrosis, and mural HIF1 α also promotes fibrosis-related gene expression and pro-fibrotic phenotypes^{3, 7, 8, 9}. In case of HIF2 α , it has been shown that HIF2 α triggers fibrosis in lung, eye, liver and kidney^{10, 11, 12, 13}, but the roles of HIF2 α in adipose tissue fibrosis remain elusive. Considering the potential roles of HIF1 α and HIF2 α in fibrosis, it is feasible to speculate that increasing beige adipocytes in adipocyte HIF α deficient models may lead to suppression of adipose tissue fibrosis. According to the reviewer's suggestion, we analyzed expression profiles of fibrosis-related genes in iWAT of WT and HIF2 α AKO mice. Similar to a previous report¹, cold stimuli inhibited overall fibrosis-related genes, whereas there were no significant differences between the genotypes (Reviewer's only Figure 1). Thus, it is likely that HIF2 α -dependent beige adipocyte regulation might not be involved in the suppression of fibrosis upon cold exposure. However, we cannot exclude the possibility that HIF2 α -miR-3085-3p axis could regulate adipose tissue fibrosis in other conditions such as CL-316,243 treatment, re-warming, and obesity. It would be interesting to investigate the roles of adipocyte HIF2 α in adipose tissue fibrosis on physiological and/or pathophysiological situation for future studies.

Reviewer's only Figure 1

a, Fragments Per Kilobase of transcript per Million (FPKM) of RNA-seq. from iWAT of WT and HIF2 α AKO mice upon thermoneutral (TN, 7 d) and cold (3 d) (GSE179385). **b**, mRNA levels in iWAT of WT and HIF2 α AKO mice upon TN (7 d) and cold (3 d). Data are expressed as the mean \pm SEM. * $P < 0.05$, ** $P < 0.01$, *** $P < 0.001$ by two-way ANOVA followed by Holm-Sidak's multiple comparisons test.

2. The authors wish to mention if the role of miR-3085 is selective to beige adipocytes in the inguinal WAT or also in iBAT.

We appreciate this comment. Our previous manuscript raised the possibility that HIF2 α would play thermoregulatory roles in brown adipocytes. According to the reviewer's comment, we analyzed expression levels of miR-3085-3p and *Prkaca* in BAT of HIF2 α BKO mice to test HIF2 α -miR-3085-3p-PKA $C\alpha$ axis. As shown in new Supplementary Fig. 8b, c, cold stimuli promote miR-3085-3p level, and cold-induced miR-3085-3p expression was downregulated in BAT of HIF2 α BKO, leading to an increase of *Prkaca* expression. Similar to beige adipocytes (Supplementary Fig. 8d, e), HIF2 α also promoted miR-3085-3p expression and suppressed *Prkaca* expression in brown adipocytes (new Supplementary Fig. 8f-i). And then, we tested the thermoregulatory roles of miR-3085-3p in brown adipocytes. Transfection of miR-3085-3p mimic in brown adipocytes attenuated *Prkaca*, PKA signaling, and PKA-induced thermogenic gene expression (new Supplementary Fig. 8l, m). Furthermore, we directly injected miR-3085-3p mimic to BAT to evaluate *in vivo* functions of miR-3085-3p in BAT using liposome-based transfection method (new Supplementary Fig. 8o). After 3 days of cold exposure following miR-3085-3p mimic transfection, BAT was examined. As shown in new Supplementary Fig. 8p, q, miR-3085-3p suppressed the levels of *Prkaca* and thermogenic genes in BAT. These data suggest that miR-3085-3p regulates thermogenic functions of BAT as well as iWAT. We described these in the revised manuscript (p.12, line 235-240, 253-256)

Reviewer #2 (Remarks to the Author):

In this manuscript Ji Seul Han et al. show that hypoxia-inducible factors -1 and -2 (HIF1 and HIF2) - are induced in inguinal WAT (iWAT) and brown adipose tissue (BAT) upon cold exposure. This compromises the expression of certain genes involved in thermogenesis and therefore loss of HIF1 α and/or HIF2 α in adipocytes (or BAT-specifically) augments surface body and rectal temperatures. Authors proposed a major role of HIF2 in this anti-thermogenic program. Indeed HIF2 α induces miR-3085-3p, which in turns represses PKAC α and therefore a protein kinase A (PKA)-dependent thermogenic/mitochondrial program in adipose tissue. Data are interesting but authors should address the following points.

Major comments:

1.- Authors show that mice treated with the HIF inhibitor YC-1 increased rectal temperature concomitant with an increased expression of thermogenic genes of iWAT upon cold exposure. YC-1 has the potential to inhibit both HIF1 and HIF2 expression (PMID: 19074848). Therefore authors should use the HIF2 specific inhibitors such as PT2385 or PT2399 (PMID: 27595394; PMID: 27595393; PMID: 31999648) to assess (i) whether this inhibitor prevents HIF2 protein accumulation in iWAT and BAT of cold-exposed mice and (ii) whether it increases rectal temperature and surface body temperature, iWAT expression of thermogenic genes as well as miR-3085-3p expression. This suggested experiment might broaden the knowledge about potential side effects of this HIF2 α inhibitor primarily oriented to treat Vhl-deficient renal cell carcinoma. Moreover authors should show whether YC-1 treatment inhibits iWAT and BAT HIF1 α and HIF2 α protein expression and conversely whether DMOG induces the expression of these two isoforms.

We appreciate this critique. Following the reviewer's suggestion, we tested the effect of HIF2 α -specific inhibitor PT2385 on thermogenic activity. PT2385 cocktail (10 mg/kg, 10% Ethanol, 40% PEG300, 5% Tween-80, and 45% saline) was orally administered twice a day from 1 day before cold exposure to 3 days after cold exposure. As shown in new Supplementary Fig. 4c, d, PT2385 selectively downregulated the level of HIF2 α protein in both iWAT and BAT. When we tested the effects of YC-1 and DMOG on HIF α protein in iWAT and BAT, YC-1 reduced levels of HIF1 α and HIF2 α protein in both iWAT and BAT (new Supplementary Fig. 4a, b). In contrast, DMOG elevated the levels of HIF1 α and HIF2 α protein in both iWAT and BAT (new Supplementary Fig. 4e, f). In order to address HIF2 α -dependent thermogenic regulation, we measured rectal temperature and surface body temperature upon cold stimuli. PT2385-treated mice exhibited cold-resistant phenotypes (new Fig. 2j and new Supplementary Fig. 4g). In addition, thermos

genic gene expression and beige adipocyte generation were upregulated by PT2385 treatment in iWAT (new Fig. 2k, l). Also, to examine HIF2 α -dependent PKA C α regulation, the levels of miR-3085-3p and *Prkaca* were measured. As shown in new Fig. 6g, h, PT2385 treatment suppressed miR-3085-3p expression, leading to an increase of *Prkaca* expression in iWAT upon cold exposure. As the reviewer mentioned, HIF2 α inhibitor has additional effects on metabolism in multiple organs besides VHL-deficient renal cell carcinoma^{14, 15, 16}. Although we cannot exclude that *in vivo* PT2385 treatment could play systemic roles in the regulation of thermogenesis besides BAT and iWAT, our pharmacological and genetic data suggest that adipocyte HIF2 α -dependent PKA regulation is crucial for the regulation of thermogenesis. These results were addressed in the revised manuscript (p.7,8,12 line 140-142, 145-147, 237-240).

2.- Authors claim that the HIF2 α -miR-3085-3p axis is central to reduce the expression of thermogenic genes as well as mitochondrial activity through the repression of PKAC α . This conclusion is drawn based on the use of a PKA inhibitor H89. Authors should use another approach with - for example - shRNA or siRNA against PKAC α in HIF2 α -deficient and control beige adipocytes and - if possible - after *in vivo* local administration in iWAT as authors do with miR-3085-3p mimic.

Thanks for this suggestion. According to the comment, we performed *Prkaca* knockdown experiments to examine PKA-dependent thermogenic regulation by HIF2 α in beige adipocytes and iWAT. Similar to H89 treatment (new Supplementary Fig. 7f), *Prkaca* knockdown attenuated thermogenic gene expression in beige adipocytes of HIF2 α AKO (new Supplementary Fig. 7g). Moreover, enhanced mitochondrial respiration in HIF2 α deficient beige adipocytes was impaired by *Prkaca* knockdown (new Supplementary Fig. 7h). We also tested *in vivo* effects of *Prkaca* knockdown in iWAT via liposome-based transfection method. Local delivery of si*Prkaca* in iWAT downregulated UCP1 and OXPHOS complexes, along with reduced beige adipocyte formation in HIF2 α AKO mice upon cold stimuli (new Fig. 5k, l). Together, these results suggest that HIF2 α -PKA C α axis would mainly regulate thermogenic execution in beige adipocytes. These data were described in the revised manuscript (p.11, line 212-218).

3.- Authors claim that HIF1 α AKO mice show a thermogenic phenotype independently of miR-3085-3p. However authors should assess whether HIF1 α AKO mice do not have elevated PKAC α expression but still can show altered mitochondrial complexes expression in iWAT after cold exposure.

The points are well taken. According to this suggestion, we investigated expression levels of mitochondrial genes and OXPHOS complexes in iWAT of HIF1 α AKO. As shown in Supplementary Fig. 7c, d, the expression of mitochondrial genes and OXPHOS complexes was upregulated in iWAT of HIF1 α AKO upon cold exposure. However, PKA C α expression was not altered by HIF1 α in beige adipocytes and iWAT (Fig. 4e, Supplementary Fig. 6h, k, and new Supplementary Fig. 6i). With these data, we affirmed that HIF1 α could regulate thermogenic functions, probably, via PKA C α -independent manner. We included these results in the revised manuscript (p.10, line 205-206).

4.- HIF1 α and HIF2 deficient mice in brown adipocytes (HIF1BKO and HIF2 α BKO) show increased body temperature but without major changes in the RNA expression of the thermogenic genes analyzed especially in HIF1 α BKO. These data can lead to consider that metabolic changes described in iWAT might not be central to explain the increased mouse temperature? Or alternatively, do these HIF1BKO and HIF2 α BKO mice have increased expression of thermogenic genes and PKAC α in iWAT possibly reflecting a possible interplay between BAT and iWAT?

We are grateful for intriguing suggestions. As the reviewer pointed out, both HIF1 α BKO and HIF2 α BKO mice were cold tolerant without significant changes in thermogenic gene expression. Since expression levels of thermogenic genes were comparable in iWAT of WT, HIF1 α BKO, and HIF2 α BKO mice (new Supplementary Fig. 3f), we assumed that there might be BAT own thermogenic regulatory mechanisms rather than interplay with iWAT. We found that miR-3085-3p expression was increased in BAT as well as iWAT upon cold exposure (new Supplementary Fig. 8b). Moreover, miR-3085-3p expression was downregulated in BAT of HIF2 α BKO, accompanied with an increase of *Prkaca* expression (new Supplementary Fig. 8b, c and Review's only Figure 2a). As brown adipocytes also showed HIF2 α -miR-3085-3p-PKA C α axis, (new Supplementary Fig. 8f-i, m), it seems

that PKA C α regulation would be an important process in BAT of HIF2 α BKO model. In HIF1 α BKO mice, *Prkaca* expression was not altered in BAT upon cold exposure (Review's only Figure 2a). Although histological differences were not largely observed at 3 days of cold exposure in BAT of WT, HIF1 α BKO, and HIF2 α BKO (Supplementary Fig. 3c, d), lipid droplets were relatively smaller in BAT of brown adipocyte HIF α deficient models than that of WT at 6 hours of cold exposure (Review's only Figure 2b, c). These data raise the possibility that HIF1 α deficiency in brown adipocytes might activate thermogenic and/or lipid catabolic pathways without regulation of thermogenic gene expression in BAT. It has been reported that thermogenic activity could be reinforced via modifications and/or modulation of UCP1 regardless of the transcriptional changes^{17, 18, 19}. Additionally, accumulating evidence has suggested that there are several processes of UCP1-independent thermogenesis^{20, 21, 22}. In future study, it will be intriguing to investigate how HIF1 α regulates thermogenic function in BAT. We described above data in the revised manuscript (p.7, 12, line 137-139, 235-240).

Reviewer's only Figure 2

a, *Prkaca* level in BAT from WT (n=8), HIF1 α BKO (n=7), and HIF2 α BKO (n=5) mice upon cold exposure (3 d). **b,c**, Representative images of H&E staining of BAT from WT, **(b)** HIF1 α BKO, and **(c)** HIF2 α BKO upon cold exposure (6 h). Scale bars, 50 μ m. Data are expressed as the mean \pm SEM. **P < 0.01 by one-way ANOVA followed by Dunnett's multiple comparison test. n.s., not significant

5.- In page 19 (Discussion section) authors mention "Furthermore, it has been reported that HIF1 α inhibitor PX-478 and HIF2 α inhibitors PT2385 and PT2399 prevent diet-induced obesity and induce thermogenic gene expression in adipose tissues 50, 51, 52". Authors should provide more details about the role of HIF2 α in thermogenesis shown in these references since in one of them (PMID: 29035368) its role is related to intestinal HIF2 α activity and not locally in the adipose tissue as shown in this manuscript. Moreover, authors should cite and discuss the following two references related to the role of HIF signaling in thermogenesis (PMID: 27738746; PMID: 34329568) especially the one in which thermogenesis is impaired upon HIF1 α expression in brown adipose tissue (PMID: 27738746).

Thanks for this critique. As the reviewer suggested, we included detailed description of the references^{14, 15, 16, 23} and addressed *in vivo* roles of PT2385 and intestinal HIF2 α in the revised manuscript. Moreover, we provided additional references^{16, 24} and discussed roles of HIF signaling in thermogenesis. The text was modified in the discussion section (p.18, line 370-380).

Minor comment:

1.- In page 7 (Results section) authors mention "Simultaneously, the levels of HIF1 α and HIF2 α proteins gradually increased during cold exposure in iWAT and BAT (Fig. 1c, d), but not in eWAT (Supplementary fig. 1b)". However, a modest HIF2 α induction was detected in eWAT upon cold exposure. Therefore, authors should change this sentence accordingly.

Thanks for this comment. We agree with the reviewer's point and revised the description (p.6, line 105-106)

References

1. Hasegawa Y, *et al.* Repression of Adipose Tissue Fibrosis through a PRDM16-GTF2IRD1 Complex Improves Systemic Glucose Homeostasis. *Cell Metab* **27**, 180-194 e186 (2018).
2. Li Y, *et al.* Comparative Transcriptome Profiling of Cold Exposure and beta3-AR Agonist CL316,243-Induced Browning of White Fat. *Front Physiol* **12**, 667698 (2021).
3. Wang W, *et al.* A PRDM16-Driven Metabolic Signal from Adipocytes Regulates Precursor Cell Fate. *Cell Metab* **30**, 174-189 e175 (2019).
4. McDonald ME, Li C, Bian H, Smith BD, Layne MD, Farmer SR. Myocardin-related transcription factor A regulates conversion of progenitors to beige adipocytes. *Cell* **160**, 105-118 (2015).
5. Koncarevic A, *et al.* A novel therapeutic approach to treating obesity through modulation of TGFbeta signaling. *Endocrinology* **153**, 3133-3146 (2012).
6. Yadav H, *et al.* Protection from obesity and diabetes by blockade of TGF-beta/Smad3 signaling. *Cell Metab* **14**, 67-79 (2011).
7. Halberg N, *et al.* Hypoxia-inducible factor 1alpha induces fibrosis and insulin resistance in white adipose tissue. *Mol Cell Biol* **29**, 4467-4483 (2009).
8. Jiang C, *et al.* Disruption of hypoxia-inducible factor 1 in adipocytes improves insulin sensitivity and decreases adiposity in high-fat diet-fed mice. *Diabetes* **60**, 2484-2495 (2011).
9. Shao M, *et al.* Pathologic HIF1alpha signaling drives adipose progenitor dysfunction in obesity. *Cell Stem Cell* **28**, 685-701 e687 (2021).
10. Li P, *et al.* SIRT1 attenuates renal fibrosis by repressing HIF-2alpha. *Cell Death Discov* **7**, 59 (2021).
11. Hickey MM, *et al.* The von Hippel-Lindau Chuvash mutation promotes pulmonary hypertension and fibrosis in mice. *J Clin Invest* **120**, 827-839 (2010).
12. Hikage F, Atkins S, Kahana A, Smith TJ, Chun TH. HIF2A-LOX Pathway Promotes Fibrotic Tissue Remodeling in Thyroid-Associated Orbitopathy. *Endocrinology* **160**, 20-35 (2019).
13. Qu A, *et al.* Hypoxia-inducible transcription factor 2alpha promotes steatohepatitis through augmenting lipid accumulation, inflammation, and fibrosis. *Hepatology* **54**, 472-483 (2011).
14. Feng Z, Zou X, Chen Y, Wang H, Duan Y, Bruick RK. Modulation of HIF-2alpha PAS-B domain contributes to physiological responses. *Proc Natl Acad Sci U S A* **115**, 13240-13245 (2018).
15. Xie C, *et al.* Activation of intestinal hypoxia-inducible factor 2alpha during obesity contributes to hepatic steatosis. *Nat Med* **23**, 1298-1308 (2017).
16. Wu Q, *et al.* Intestinal hypoxia-inducible factor 2alpha regulates lactate levels to shape the gut microbiome and alter thermogenesis. *Cell Metab* **33**, 1988-2003 e1987 (2021).
17. Chouchani ET, *et al.* Mitochondrial ROS regulate thermogenic energy expenditure and sulfenylation of UCP1. *Nature* **532**, 112-116 (2016).
18. Fedorenko A, Lishko PV, Kirichok Y. Mechanism of fatty-acid-dependent UCP1 uncoupling in brown fat mitochondria. *Cell* **151**, 400-413 (2012).
19. Shi M, *et al.* AIDA directly connects sympathetic innervation to adaptive thermogenesis by UCP1. *Nat Cell Biol* **23**, 268-277 (2021).
20. Cohen P, Kajimura S. The cellular and functional complexity of thermogenic fat. *Nat Rev Mol Cell Biol* **22**, 393-409 (2021).
21. Ikeda K, *et al.* UCP1-independent signaling involving SERCA2b-mediated calcium cycling regulates beige fat thermogenesis and systemic glucose homeostasis. *Nat Med* **23**, 1454-1465 (2017).
22. Kazak L, *et al.* A creatine-driven substrate cycle enhances energy expenditure and thermogenesis in beige fat. *Cell* **163**, 643-655 (2015).
23. Sun K, Halberg N, Khan M, Magalang UJ, Scherer PE. Selective inhibition of hypoxia-inducible factor 1alpha ameliorates adipose tissue dysfunction. *Mol Cell Biol* **33**, 904-917 (2013).
24. Jun JC, *et al.* Adipose HIF-1alpha causes obesity by suppressing brown adipose tissue thermogenesis. *J Mol Med (Berl)* **95**, 287-297 (2017).

REVIEWERS' COMMENTS

Reviewer #1 (Remarks to the Author):

The authors provided sufficient evidence that addressed the reviewer's comments. No further concerns from this reviewer.

Reviewer #2 (Remarks to the Author):

Authors have addressed satisfactorily all my comments. Just a minor comment, authors mention, "In BAT, histological changes upon cold exposure were not significantly different between the genotypes (Supplementary fig. 3c, d),.....". However - based on the data shown exclusively in the response letter - authors might consider to include in the manuscript that the size of brown adipocytes was reduced after 6 hours of cold exposure in HIF1a BKO and HIF2a BKO mice. A very brief discussion including a possible explanation of why these differences are lost after 3 days of cold exposure might be included.

Response to the reviewers' comments

manuscript NCOMMS-21-32870A

Adipocyte HIF2 α functions as a thermostat via PKA C α regulation in beige adipocytes

Reviewer #1 (Remarks to the Author):

The authors provided sufficient evidence that addressed the reviewer's comments. No further concerns from this reviewer.

Reviewer #2 (Remarks to the Author):

Authors have addressed satisfactorily all my comments. Just a minor comment, authors mention, "In BAT, histological changes upon cold exposure were not significantly different between the genotypes (Supplementary fig. 3c, d),.....". However - based on the data shown exclusively in the response letter - authors might consider to include in the manuscript that the size of brown adipocytes was reduced after 6 hours of cold exposure in HIF1 α BKO and HIF2 α BKO mice. A very brief discussion including a possible explanation of why these differences are lost after 3 days of cold exposure might be included.

Thank you for the suggestion. In the previous version of manuscript, there were incorrect histological images (old Supplementary Fig. 3c, d). We substituted the images with BAT histology of 3 days of cold exposure (new Supplementary Fig. 3d), and included BAT histology of 6 hours of cold exposure in the revised manuscript (new Supplementary Fig. 3c). Also, we described this in the text (p.7, line 136-140).